# Stimulatory and inhibitory G-protein signaling relays drive cAMP accumulation for timely metamorphosis in the chordate *Ciona*

Akiko Hozumi[1†], Nozomu M Totsuka[2†], Arata Onodera[1], Yanbin Wang[1], Mayuko Hamada[3], Akira Shiraishi[4], Honoo Satake[4], Takeo Horie[5], Kohji Hotta[2], Yasunori Sasakura[1]*

[1]Shimoda Marine Research Center, University of Tsukuba, Shimoda, Japan; [2]Department of Biosciences and Informatics, Faculty of Science and Technology, Keio University, Yokohama, Japan; [3]Ushimado Marine Institute, Okayama University, Okayama, Japan; [4]Bioorganic Research Institute, Suntory Foundation for Life Sciences, Kyoto, Japan; [5]Laboratory for Single-cell Neurobiology, Graduate School of Frontier Biosciences, Osaka University, Suita, Japan

*For correspondence:
sasakura@shimoda.tsukuba.ac.jp

†These authors contributed equally to this work

## eLife Assessment

This **important** work substantially advances our understanding of the molecular mechanisms underlying the timing of the initiation of metamorphosis of the Ciona ascidian tadpole larva. Through the combination of gene knockdown experiments and fluorescent molecular reporters the authors provide **compelling** evidence about a crosstalk between different G protein mediated signalling pathways and are able to place different signalling molecules within a signalling network. The work will be of interest to molecular, developmental and marine biologists and to scientists working on animal metamorphosis.

**Abstract** Larvae of the ascidian *Ciona* initiate metamorphosis tens of minutes after adhesion to a substratum via their adhesive organ. The gap between adhesion and metamorphosis initiation is suggested to ensure the rigidity of adhesion, allowing *Ciona* to maintain settlement after losing locomotive activity through metamorphosis. The mechanism producing the gap is unknown. Here, by combining gene functional analyses, pharmacological analyses, and live imaging, we propose that the gap represents the time required for sufficient cyclic adenosine monophosphate (cAMP) accumulation to trigger metamorphosis. Not only the Gs pathway but also the Gi and Gq pathways are involved in the initiation of metamorphosis in the downstream signaling cascade of the neurotransmitter GABA, the known initiator of *Ciona* metamorphosis. The mutual crosstalk of stimulatory and inhibitory G-proteins functions as the accelerator and brake for cAMP production, ensuring the faithful initiation of metamorphosis at an appropriate time and in the right situation.

## Introduction

Metamorphosis is a widespread feature of development that allows animals to have different functions between larval and adult stages (*Barresi and Gilbert, 2020*). As larvae become adults, the shape and various characteristics of their physiology, gene expression, behavior, and lifestyle change.

**eLife digest** Ascidians (also known as sea squirts) are small marine animals. Ascidian larvae have tadpole-like bodies and swim by beating their tail, but the adults have a vase-like shape, no tail and are attached to rocks or other solid surfaces, unable to move. The process by which an ascidian larva transforms into an adult, known as metamorphosis, starts when the larvae stick to a surface using parts of their body known as adhesive papillae.

A group of ascidians known as *Ciona* are widely studied due to the availability of molecular biology methods for understanding their genetics. Previous studies have shown that *Ciona* larvae must remain stuck to a surface for approximately 30 minutes to trigger metamorphosis. This time gap between attaching to a surface and metamorphosis is thought to be a system for ensuring an ascidian is attached securely to a suitable surface before it loses its ability to move. However, it remains unclear how *Ciona* can measure time in this situation.

To address this question, Hozumi, Totsuka et al. combined genetics, pharmacology and live imaging approaches to study *Ciona* larvae as they prepared for metamorphosis. The experiments found that *Ciona* larvae made a molecule known as cyclic adenosine monophosphate (or cAMP, for short) in their adhesive papillae once they had attached to a surface. Over time, the levels of cAMP in the papillae gradually increased, providing the larvae remained attached. When the levels of cAMP reached a threshold, *Ciona* initiated metamorphosis.

These findings demonstrate that *Ciona* larvae use cAMP levels as a timer to measure how long they have been attached to a surface. This molecule is found in all living organisms; therefore, other animals may use similar mechanisms to measure time. Researchers need to be able to artificially control the timing of biological events to efficiently culture cells and tissues in the laboratory. Therefore, the genes that regulate cAMP production in *Ciona* and other organisms may be useful targets for developing new methods for growing organisms in the laboratory.

With these changes, animals can focus on a limited number of biological activities at each stage to increase feeding, growth, dispersal, and reproduction efficiencies. This biphasic mode of life is a widely conserved and ancient trait of animals, as evidenced by its presence in groups that maintain primitive states (*Arenas-Mena, 2010*), indicating that this long-conserved feature has contributed to animals' flourishment. Therefore, to understand animal evolution, it is essential to elucidate the mechanisms underlying metamorphosis.

A key characteristic of metamorphosis is that both internal and external conditions determine its initiation. Sufficiently matured (or metamorphically competent *Hadfield et al., 2001*) larvae start metamorphosis only when they meet an appropriate external condition to be adults. Various organic and inorganic external stimuli suited to the lifestyle of adulthood trigger metamorphosis. Many marine invertebrates exhibit a benthic lifestyle at the adult stage (*Degnan and Degnan, 2010*). Their planktonic larvae have an adhesive organ that secretes adhesives and adheres to a substratum. The cues associated with the adhesion, such as the physical contact with the substratum and a chemical from organisms surrounding the adherence site, can trigger their metamorphosis. Upon starting metamorphosis, their larvae lose locomotive organs and transition into benthic adult forms. Determining the timing of metamorphosis is important for benthic animals, particularly sessile ones, to ensure their future survival and reproduction, as they become unable to relocate after metamorphosis. Therefore, sessile animals are suspected to have elaborate mechanisms to start metamorphosis only when a firm adhesion is achieved at an appropriate place. Although several reports describe this kind of phenomenon (*Williams et al., 2009*; *Say and Degnan, 2020*; *Yap et al., 2023*), the mechanism by which the external condition is turned into an internal mechanism to trigger metamorphosis at the right timing remains elusive.

Ascidians are marine invertebrate chordates that are the closest living relatives to vertebrates (*Delsuc et al., 2006*; *Satoh, 2003*; *Bourlat et al., 2006*; *Delsuc et al., 2008*). The chordate features of ascidians are best represented by their tadpole larval shape. Like vertebrate tadpoles, ascidian larvae swim by beating their tail through neuromuscular activities (*Nishino et al., 2011*; *Horie et al., 2010*). However, ascidians lose the tadpole shape during metamorphosis, and they exhibit a sessile lifestyle at the adult stage (*Karaiskou et al., 2015*; *Hotta et al., 2020*). Ascidian metamorphosis comprises

several key events, such as tail regression, body axis rotation, and adult organ growth (*Cloney, 1982*; *Figure 1A*).

Adhesion to a substratum triggers ascidian metamorphosis (*Cloney, 1982*). Ascidian larvae have adhesive organs called adhesive papillae, usually triangular protrusions at the anterior end. Adhesive papillae secrete a lectin-type mucus substance that is suspected to aid in binding to the substratum (*Zeng et al., 2019*). Moreover, adhesive papillae serve as mechanical sensors (*Wakai et al., 2021*). Although cells in the papillae are polymodal, sensing both mechanical and chemical stimuli (*Hoyer et al., 2024*), the mechanical stimulus can trigger metamorphosis without such a chemical (*Wakai et al., 2021*; *Sensui and Hirose, 2020*) under laboratory conditions. The papilla is a neuronal organ that includes epidermal neurons positive for vesicular glutamate transporter (VGLUT) and GABA, which innervate axons toward the sensory vesicle (larval brain) (*Takamura, 1998*; *Horie et al., 2008*; *Brown et al., 2005*; *Chacha et al., 2022*). The papilla neurons (PNs) are responsible for initiating metamorphosis (*Wakai et al., 2021*; *Nakayama-Ishimura et al., 2009*; *Sakamoto et al., 2022*; *Johnson et al., 2023*). Moreover, our recent study suggests the involvement of a mechanoreceptor channel (the TRP channel) in the responsiveness to mechanical stimuli (*Sakamoto et al., 2022*). However, a simple physical adhesion is not sufficient to trigger metamorphosis. In our study using the model ascidian *Ciona intestinalis* Type A (it has been proposed that this species be renamed *C. robusta Caputi et al., 2007*; *Pennati et al., 2015*; *Brunetti et al., 2015*, and hereafter we call it *Ciona*), continuous adhesion for about 30 min is necessary before starting metamorphosis (*Matsunobu and Sasakura, 2015*). When larvae detach before reaching the critical period, they must adhere for 30 min again for metamorphosis. Therefore, it is thought that ascidian larvae somehow sense the duration of adhesion and initiate metamorphosis only when adhesion is firm enough to be maintained for the required period. *Ciona* larvae continue tail beating during adhesion to push their body to the substratum. We showed that the strength of force generated by this swimming activity influences the timing of metamorphosis initiation (*Sakamoto et al., 2022*). In this phenomenon, the abolishment of swimming activity elongates the time from settlement until metamorphosis is initiated. The mechanisms for measuring the duration of adhesion and the strength of the force generated by adhesion are unknown.

To elucidate the mechanisms triggering ascidian metamorphosis, the signaling pathways must be characterized. Many studies have discovered signaling molecules as possible inducers of ascidian metamorphosis (*Hoyer et al., 2024*; *Patricolo et al., 2001*; *Arnold et al., 1997*; *Coniglio et al., 1998*; *Eri et al., 1999*; *Kimura et al., 2003*; *Zega et al., 2005*; *Patricolo et al., 1981*). These molecules include neurotransmitters, suggesting that the transmission of excitation in the nervous system, starting from the adhesive papillae, is crucial for metamorphosis. Recently, our group reported that the inhibitory neurotransmitter GABA plays a pivotal role in initiating *Ciona* metamorphosis (*Hozumi et al., 2020*). Knockout and knockdown of the genes encoding the enzyme synthesizing GABA called glutamate decarboxylase (GAD), vesicular inhibitory amino acid transporter (VIAAT or VGAT), and metabotropic GABA receptor (GABABR) resulted in the perturbation of metamorphosis. Moreover, GABA administration induces metamorphosis without adhesion. Our previous studies demonstrated that GABA induces the secretion of gonadotropin-releasing hormone (GnRH) (*Hozumi et al., 2020*; *Kamiya et al., 2014*); however, it remains unknown how the inhibitory neurotransmitter activates neuronal functions for initiating metamorphosis.

In this study, we addressed the characterization of the downstream cascade stimulated by GABA. Because GABABR is a G-protein-coupled receptor (GPCR) (*Shaye et al., 2021*), we searched for trimeric G-proteins necessary for metamorphosis. These G-proteins are heterotrimers of an αβ and γ subunit (*Alberts et al., 2017*). Upon ligand binding, GPCR exchanges GDP of the α subunit to GTP, then the GTP-bound α subunit and the βγ complex are released from the GPCR. Although both the GTP-bound α subunit and the βγ complex have activities, the α subunit mainly determines the reaction specific to the G-protein type. Gαs and Gαi, respectively, activate and inhibit cyclic adenosine monophosphate (cAMP) synthesis. In contrast, Gαq increases intracellular $Ca^{2+}$ ion concentration by promoting its secretion from the endoplasmic reticulum. We found that three G-proteins are activated in the downstream GABA cascade, resulting in cAMP elevation. Because the signaling includes stimulating and inhibiting cAMP synthesis, *Ciona* initiates metamorphosis only when a sufficient quantity of cAMP is accumulated due to sustained adhesion. Our results revealed the ingenious mechanism that permits *Ciona* to start metamorphosis only when achieving an appropriate adhesion firm enough to relinquish swimming ability and commence sessile adult life.

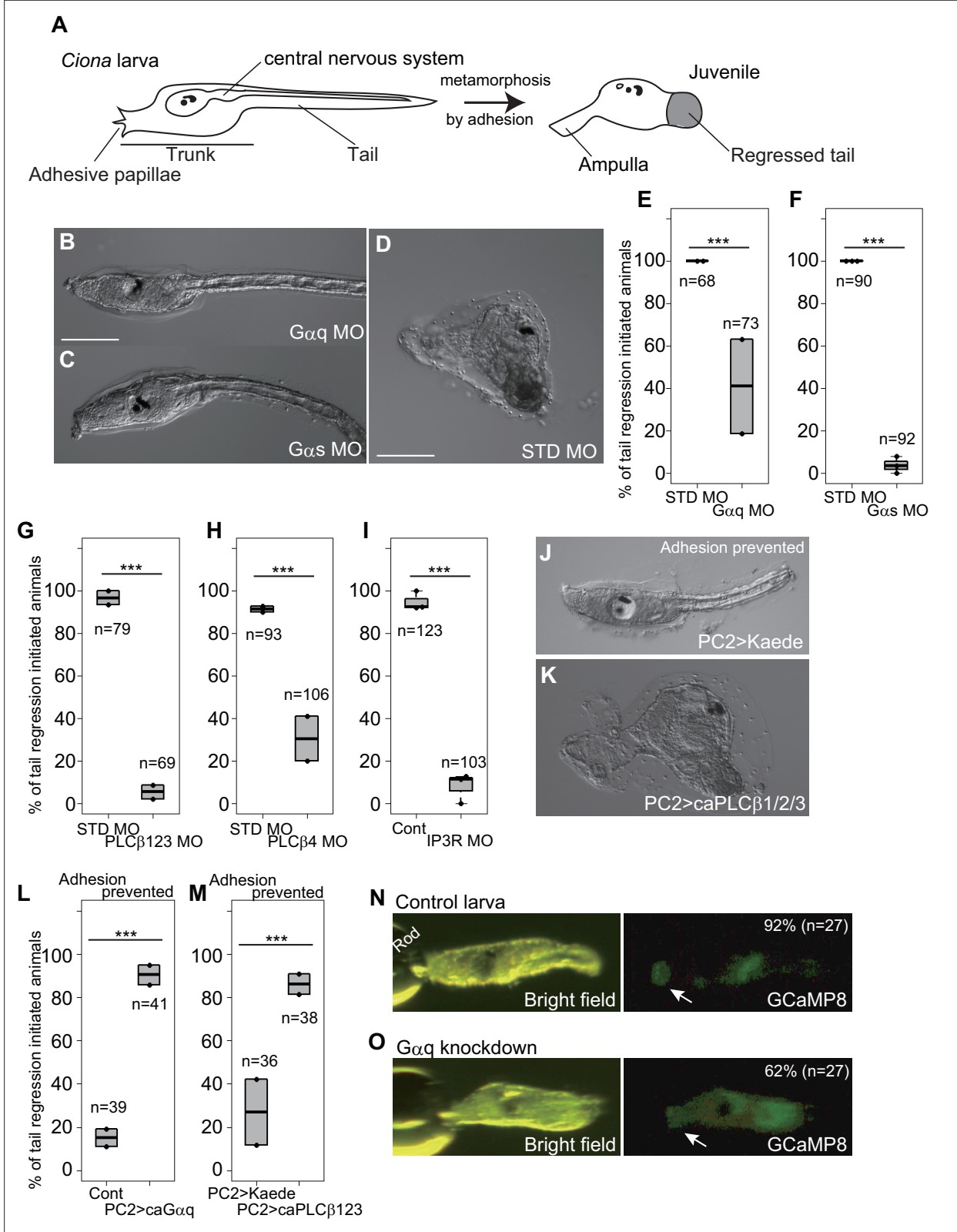

**Figure 1.** Gq and Gs pathways are required for *Ciona* metamorphosis. (**A**) Schematic illustration of *Ciona* metamorphosis. (**B**) A larva knocked down with *Gαq* using the antisense morpholino oligonucleotide (MO). Metamorphosis did not initiate at 2 days post-fertilization (2 dpf). Scale bar, 100 µm. (**C**) A *Gαs* knockdown larva. (**D**) A control animal injected with the standard (STD) MO. Metamorphosis initiated, as indicated by the completion of tail regression. Scale bar, 100 µm. (**E**) Effect of *Gαq* knockdown on the percentage of metamorphosis initiation (indicated by the initiation of tail regression),

*Figure 1 continued on next page*

*Figure 1 continued*

shown as box-and-whisker plots. Dots indicate experiment replicates. ***p < 0 .001 (Fisher's exact test). *n*, number of examined larvae in total. (**F**) Effect of *Gαs* knockdown. (**G–I**) Effects of *PLCβ1/2/3*, *PLCβ4*, and *IP3R* knockdowns. (**J–M**) Results of the experiments where adhesion was prevented by tail amputation and laying larvae on agar-coated plates. (**J**) A control larva overexpressing the *Kaede* reporter in the entire nervous system using the *PC2 cis* element. (**K**) A larva overexpressing constitutive active (ca)*PLCβ1/2/3*. (**L**) Effect of ca*Gαq* overexpression. (**M**) Effect of ca*PLCβ1/2/3* overexpression. (**N, O**) Live imaging of $Ca^{2+}$ transient, monitored by injecting *GCaMP8* mRNA. (**N**) Snapshot of the control larva injected with *GCaMP8* mRNA. An increase in the fluorescence intensity in the adhesive papillae (arrow) was observed. Rod, the position of the glass rod used to stimulate papillae. The percentage and number exhibit the rate of animals showing $Ca^{2+}$ transient in the papillae. (**O**) A larva injected with *Gαq* MO plus *GCaMP8* mRNA. An increase in the fluorescence intensity in the papillae (arrow) was not observed even though the brightness of the whole body was raised.

The online version of this article includes the following figure supplement(s) for figure 1:

**Figure supplement 1.** *Ciona* genes encoding the G alpha subunit.

**Figure supplement 2.** *Ciona* phospholipase β proteins.

**Figure supplement 3.** *Ciona* phosphodiesterase (PDE) repertoire and expression of Gq and Gs pathway genes in the adhesive papillae.

## Results

### Characterization of G-proteins necessary for metamorphosis initiation

A previous study (*Prasobh and Manoj, 2009*) identified ten genes encoding Gα proteins from the *Ciona* genome, and we found nine gene models corresponding to these proteins in the latest genome assembly (*Satou et al., 2022*; *Supplementary file 1*). Gα proteins are classified into four major groups (Gαs, Gαi, Gαq, and Gα12/13), which have distinct functional properties (*Preininger and Hamm, 2004*). *Ciona* has one gene encoding an unambiguous ortholog in each of the four groups (*Figure 1—figure supplement 1A*). Moreover, a previous study (*Prasobh and Manoj, 2009*) suggested the presence of another Gαq protein in the *Ciona* genome. Our phylogenetic tree suggested that Gαq (Gαq_Chr11) is orthologous to human GNA15 (*Figure 1—figure supplement 1A*).

The remaining four *Ciona* genes (their gene models are KY21.Chr2.875, KY21.Chr4.943, KY21.Chr9.455, and KY21.Chr8.580) encode divergent Gα proteins. The Gα protein encoded by KY21.Chr8.580 has truncation at its N-terminal part, and we omitted this protein from the phylogenetic analysis. Our phylogenetic analyses showed that the remaining divergent Gα proteins have a strong affinity to the Gαi/o family proteins (*Figure 1—figure supplement 1A*). Moreover, the amino acid residues at the C-terminal end of these proteins are more similar to human Gαi than the other Gα protein families (*Figure 1—figure supplement 1B*). The C-terminal residues, particularly glycine at position 3, are essential for determining the Gi partner of GPCR (*Conklin et al., 1993*). We tentatively name these proteins dvGαi_Chr2, dvGαi_Chr4, and dvGαi_Chr9. Gαi/o family proteins have inhibitory activity that suppresses cAMP production (*Syrovatkina et al., 2016*).

To verify the possible contribution of G-proteins to the initiation of metamorphosis, we quantified the expression level of the Gα genes in the adhesive papillae using RNA-seq surrounding this region (*Figure 1—figure supplement 1C* and *Supplementary file 1*). Typical *Gαq* and three *Gαi* genes exhibited more abundant expression in the papillae, followed by *Gαs* and *Gα12/13*. Therefore, these Gα proteins were chosen as candidates for the protein(s) regulating metamorphosis initiation. Among them, we prioritized examining the Gα proteins having an excitatory function (Gαq and Gαs) rather than an inhibitory role, since previous studies suggested that excitatory events such as $Ca^{2+}$ transient and neuropeptide secretion occur when *Ciona* metamorphose (*Wakai et al., 2021*; *Hozumi et al., 2020*). The primary function of Gα12/13 is the activation of Rho-mediated actin dynamics. Although this regulation is essential for tail regression (*Yamaji et al., 2020*), this event occurs after metamorphosis initiation. Therefore, we omitted this gene from further analysis in this study.

We knocked down the *Gαq* and *Gαs* genes using antisense morpholino oligonucleotides (MOs). We found that both of of these genes are necessary for metamorphosis. The morphants of these genes showed a reduced rate of metamorphosis initiation even though they were allowed to adhere to the base of culture dishes, which was sufficient for control larvae to initiate metamorphosis (*Figure 1A–F*). *Gαs* MO exhibited almost complete perturbation of metamorphosis, while the effect of *Gαq* MO was milder (*Figure 1E, F*).

Gαq activates phospholipase C beta (PLCβ) to produce inositol triphosphate (IP3) (*Harden et al., 2011*). IP3 is received by its receptor on the endoplasmic reticulum that releases calcium ion ($Ca^{2+}$). The *Ciona* genome encodes two PLCβ (PLCβ1/2/3, PLCβ4) and one IP3 receptor (IP3R) (*Figure 1—figure*

supplement 2 and *Supplementary file 1*). We knocked down the *PLCβ1/2/3*, *PLCβ4*, and *IP3R* genes. The knockdown larvae of these three genes failed to start metamorphosis (*Figure 1G–I*). The effect of *PLCβ4* MO was weaker than those of the other MOs, suggesting that this PLC plays an auxiliary role. These results suggest that Gαq initiates metamorphosis by the conventional $Ca^{2+}$ pathway mediated by PLCβ and IP3/IP3R. To confirm this, we overexpressed constitutively active forms of Gαq (caGαq) (*Oki et al., 2005*) and of caPLCβ1/2/3 (*Charpentier et al., 2014*) throughout the entire nervous system with the *cis* element of the gene encoding prohormone convertase 2 (PC2) (*Yokoyama et al., 2014*; *Osugi et al., 2020*; *Osugi et al., 2017*). When the microinjected animals reached the larval stage, they were cultured on agar-coated dishes after tail amputation to prevent adhesion. In this condition, the control larvae rarely started metamorphosis because of the absence of adhesion (*Figure 1J*). In contrast, ca*Gαq*- or ca*PLCβ1/2/3*-overexpressed larvae initiated metamorphosis without adhesion (*Figure 1K–M*).

Adhesive papillae exhibit a $Ca^{2+}$ increase soon after sensing an adhesive stimulus (*Wakai et al., 2021*; *Sakamoto et al., 2022*). We examined whether this $Ca^{2+}$ transient is dependent on the Gq pathway. Compared to controls, significantly fewer larvae injected with *Gαq* MO plus *GCaMP8* mRNA exhibited GCaMP8 fluorescence elevation in the adhesive papilla upon stimulation (*Figure 1N, O*). This result suggests that the Gq pathway is activated upon adhesion to cause a $Ca^{2+}$ transient in the adhesive papilla.

## The Gs pathway initiates metamorphosis by activating cAMP synthesis

The involvement of Gs in metamorphosis was confirmed by the overexpression of a constitutively active form of Gαs (caGαs) (*Kelly et al., 2007*) in the nervous system. This overexpression resulted in the enhanced initiation of metamorphosis without adhesion (*Figure 2A–C*). The Gs pathway activates adenylate cyclase (AC) to produce cAMP (*Neves et al., 2002*). We previously reported that cAMP can induce metamorphosis (*Kamiya et al., 2014*). Because externally added cAMP is not a strong inducer of metamorphosis, we attempted to confirm this hypothesis through another experiment. Theophylline increases cAMP by inhibiting the cAMP-degrading enzyme phosphodiesterase (PDE) (*Essayan, 2001*). We treated wild-type larvae with theophylline after tail amputation, and we observed that most theophylline-treated larvae completed tail regression without adhesion (*Figure 2D–F*). Theophylline has several target proteins in addition to PDE (*Daly et al., 1987*). To further confirm that cAMP is responsible for the initiation of metamorphosis, we overexpressed photo-activating AC (bPAC) (*Stierl et al., 2011*) in the nervous system. The *bPAC*-overexpressed larvae regressed their tails without adhesion, suggesting that a cAMP increase triggers metamorphosis (*Figure 2G*).

Using bPAC, we addressed whether enhanced cAMP production facilitates the initiation of metamorphosis. At 24 hours post-fertilization (hpf), only a few animals in both the *bPAC*-overexpressed and control groups initiated metamorphosis because 24 hpf is somewhat too early for *Ciona* larvae to be metamorphically competent (*Matsunobu and Sasakura, 2015*; *Figure 2H*). Even in this condition, the *bPAC*-overexpressed group exhibited a statistically higher rate of metamorphosis-initiated larvae. The proportion of animals initiating metamorphosis increased over successive time points, with the *bPAC*-overexpressed group consistently showing a statistically higher rate of metamorphosis compared to controls. These results support the idea that cAMP plays a role as a timer for *Ciona* metamorphosis; accumulation of this molecule to exceed a threshold could trigger metamorphosis. Because many of the *bPAC*-overexpressed larvae did not initiate metamorphosis at 24 hpf, similar to control larvae, cAMP accumulation does not alter the timing of acquiring metamorphic competence.

## Gq–Gs pathways work in the adhesive papillae for metamorphosis

The above results showed that activation of the Gq and Gs pathways is the key events in initiating metamorphosis. Gαq is necessary to induce $Ca^{2+}$ transients in the adhesive papillae, suggesting that the Gq pathway functions in this region. If both Gq and Gs function in the papillae, they should be expressed in the adhesive papilla. The transcriptome analysis of the larval papilla region showed that the genes encoding Gq and Gs pathway proteins are expressed in this region (*Figure 1—figure supplement 1C*, *Figure 1—figure supplement 3*) and (*Supplementary file 1*). Therefore, Gq and Gs could function in the papilla to initiate metamorphosis.

As shown above, theophylline induced metamorphosis without settlement. However, when the papillae were removed from larvae, the average rate of metamorphosis induction by theophylline

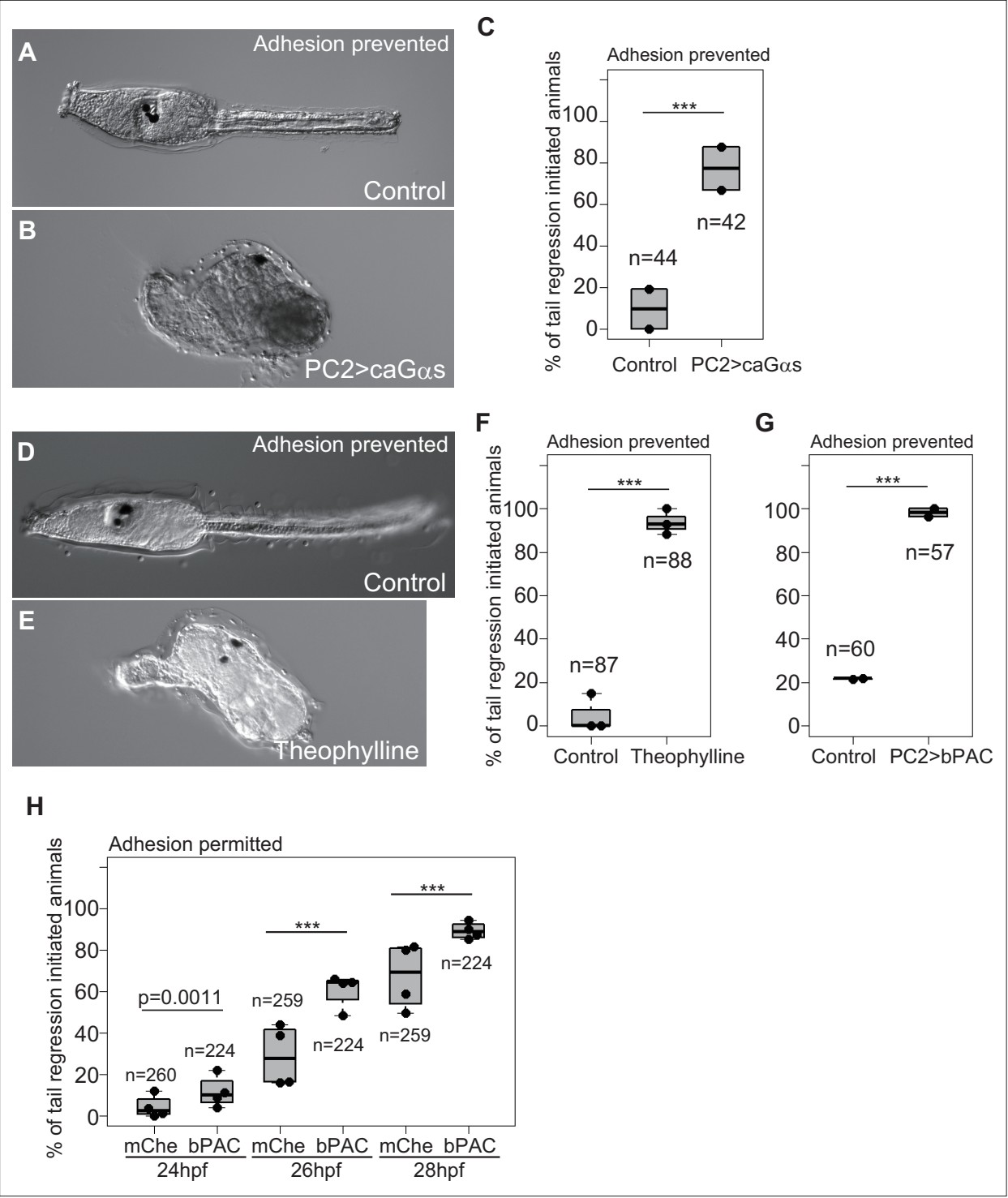

**Figure 2.** cAMP initiates *Ciona* metamorphosis. (**A**) An adhesion-prevented control (uninjected) larva at 2 dpf. Metamorphosis did not initiate.
(**B**) An animal with adhesion prevention and overexpression of caGαs in the entire nervous system. Metamorphosis initiated. (**C**) Effect of caGαs
overexpression. ***, p<0.001 (Fisher's exact test). (**D**) A control (uninjected) larva. (**E**) A theophylline-treated animal. Metamorphosis initiated. (**F**) Effect
of theophylline treatment. (**G**) Effect of *bPAC* overexpression in the nervous system. (**H**) Effect of *bPAC* overexpression over time. In this experiment,
adhesion was permitted. Control groups (overexpressed with mCherry) exhibited a slower initiation of metamorphosis than bPAC-overexpressed
groups.

was reduced from 100% to 30% (*Figure 3A–C*). This suggests that cAMP elevation in the adhesive papillae is essential for starting metamorphosis. The overexpression of ca*Gαq*, ca*PLCβ1/2/3*, ca*Gαs*, and *bPAC* by the *PC2 cis* element resulted in the initiation of metamorphosis without adhesion. Like theophylline, amputation of the papillae reduced their effects in starting metamorphosis (*Figure 3D*), confirming that activation of the Gq and Gs pathways in the adhesive papillae triggers metamorphosis. Among these experiments, ca*PLCβ1/2/3* overexpression was the most sensitive to papilla amputation, suggesting that PLCβ1/2/3 acts specifically in the papillae during metamorphosis.

If the Gs pathway is activated in the adhesive papillae, a cAMP increase should be observed in this region upon adhesion. We examined this possibility using a fluorescent cAMP indicator called Pink Flamindo (PF) (*Harada et al., 2017*). After stimulation, the fluorescence of PF in the papillae temporarily decreased on average by 0.83-fold ($n = 5$) from the initial intensity, followed by a gradual increase by 0.0102 per min, to reach 1.17-fold (*Figure 3E*, *Figure 3—figure supplement 1A*). The cAMP reduction and increase, respectively, started at 35 s and 4 min 40 s after stimulation on average. Such a reduction and increase in fluorescence intensity was not observed in the adhesive papillae of the larvae that had failed to initiate metamorphosis following stimulation ($n = 4$; *Figure 3—figure supplement 1B*). Therefore, increased cAMP in the papillae serves to indicate that sufficient stimuli of adhesion have been received to induce metamorphosis. This strengthens our hypothesis that cAMP accumulation in the adhesive papillae determines the initiation of metamorphosis. Both the *Gαs* and *Gαq* knockdowns abolished the temporal cAMP decrease and reduced the rate of subsequent accumulation (*Figure 3—figure supplement 1C, D*), indicating that these two events are dependent on the Gs and Gq pathways. However, neither knockdown abolished the increase in cAMP completely (as shown by the slope scores), suggesting the presence of a mechanism producing cAMP without their activations.

## GABA in the adhesive papillae is responsible for metamorphosis

Our previous study demonstrated that GABA is the chief neurotransmitter that induces metamorphosis (*Hozumi et al., 2020*). To gain insight into the relationships between the GABA, Gq, and Gs pathways, we addressed whether GABA functions in the adhesive papillae for initiating metamorphosis, similar to Gq and Gs.

The previous studies and our transcriptome data suggest that *GAD* is expressed in the adhesive papillae (*Zega et al., 2008*; *Supplementary file 1*). Moreover, GABA-immunopositive signals have been detected in the papillae (*Brown et al., 2005*; *Zega et al., 2008*). Therefore, papillae can provide GABA to stimulate themselves. Next, we examined whether the adhesive papillae can receive GABA to initiate metamorphosis. We showed that the metabotropic GABA receptor is responsible for the initiation of metamorphosis (*Hozumi et al., 2020*). Using whole-mount in situ hybridization, the previous study did not detect the expression of two GABAB receptor (GABABR) genes in the papillae (*Zega et al., 2008*); however, our transcriptome data detected the low-level expression of three genes encoding GABABR (*Supplementary file 1*) in the papillae, suggesting that adhesive papillae could receive GABA. GABA can induce metamorphosis without adhesion (*Figure 3—figure supplement 2A*; *Hozumi et al., 2020*). However, the amputation of adhesive papillae suppressed this activity (*Figure 3—figure supplement 2B, C*), suggesting that GABA reception by the papillae is responsible for starting metamorphosis.

The larval brain is the major domain expressing *GABABR* (*Zega et al., 2008*). Therefore, it remains possible that GABA signaling in the brain stimulates the papillae in a retrograde manner to initiate metamorphosis. The connectome analyses of the larval nervous system did not suggest a nerve that inputs into the papillae from the brain (*Ryan et al., 2016*; *Ryan et al., 2018*); however, retrograde stimulation would be possible through the volume transmission of a signaling molecule. To examine this possibility, we isolated larval fragments anterior to the brain (*Figure 3—figure supplement 2D*), followed by the administration of GABA and theophylline. Through the application of these chemicals, anterior fragments exhibited increased clarity, elongation, and retraction of papillae, which are the signatures of metamorphosis (*Figure 3—figure supplement 2E–G*). Because their responses to GABA were weaker than those observed in the theophylline treatment, we further examined whether the anterior fragments can respond to GABA by monitoring the expression of GABA-responsive genes. By comparing expression levels between control and GABA-administered larvae, we made a list of the genes upregulated by GABA in the papillae (*Supplementary file 2*). Among them, we compared

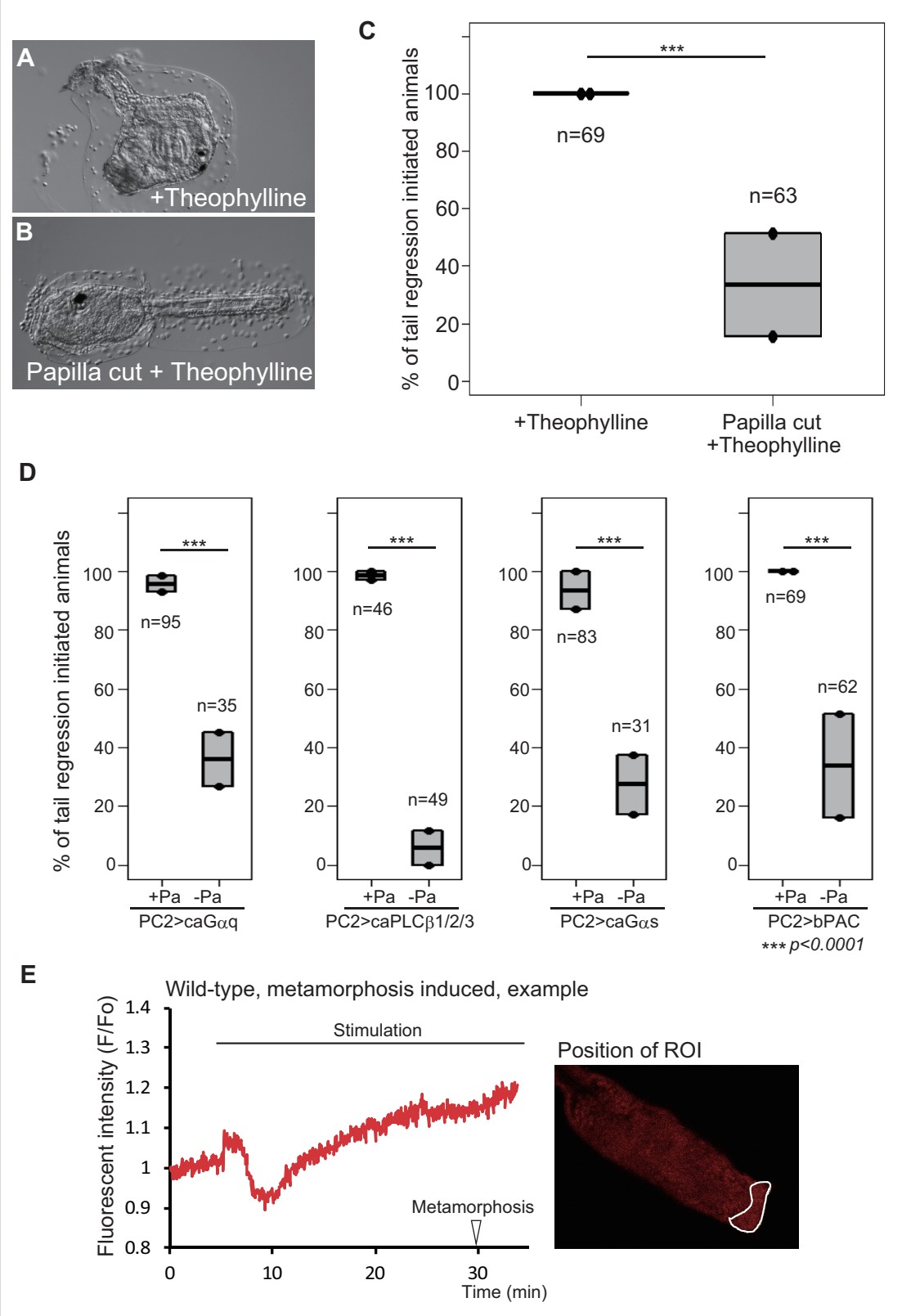

**Figure 3.** Gq and Gs pathways are activated in the adhesive papillae to initiate metamorphosis. (**A**) A theophylline-treated animal. Metamorphosis initiated. (**B**) A papilla-amputated larva treated with theophylline. Metamorphosis did not initiate. (**C**) Effect of papilla amputation on theophylline treatment. ***, p<0.001 (Fisher's exact test). (**D**) Effects of papilla amputation on ca*Gαq*, ca*PLCβ1/2/3*, ca*Gαs*, and bPAC overexpression. (**E**) Measurement of cAMP concentration, as indicated by the fluorescence of Pink Flamindo. Left, the quantification of Pink Flamindo fluorescent

*Figure 3 continued on next page*

*Figure 3 continued*

intensity is shown for a larva that initiated metamorphosis at the time indicated by an arrowhead. The time of stimulation of the adhesive papillae with a glass rod is denoted by a bar. Right, the position of the region of interest (ROI).

The online version of this article includes the following figure supplement(s) for figure 3:

**Figure supplement 1.** cAMP increase in the papillae is coupled with metamorphosis initiation.

**Figure supplement 2.** GABA stimulates the adhesive papillae to initiate metamorphosis.

the expression levels of two genes (KY21.Chr5.240 and KY21.Chr8.489) between GABA-administered and control anterior fragments. GABA-treated anterior fragments expressed these genes at levels at least three times higher than controls (*Figure 3—figure supplement 2H*). We concluded that GABA is secreted from and stimulates the papillae to start metamorphosis.

## Gs/cAMP is downstream of the Gq pathway

Our next question is: What are the upstream or downstream relationships between GABA, Gq/Ca$^{2+}$, and Gs/cAMP in the adhesive papillae? We first examined the relationships between the Gq and Gs pathways. Theophylline ameliorated metamorphosis-failed phenotypes in *Gαq*, *PLCβ*, and *IP3R* knockdowns (*Figure 4A–E*). Moreover, ca*Gαs* overexpression in the nervous system significantly, although not completely, ameliorated the effect of *Gαq* MO (*Figure 4F*). These results suggest that the Gq pathway is upstream of the Gs/cAMP pathway. *Gαs* knockdown larvae exhibited Ca$^{2+}$ transients in the adhesive papillae upon stimulation (*Figure 4G*). Because this Ca$^{2+}$ transient is Gq dependent (*Figure 1O*), this result confirms that the Gs pathway is downstream of the Gq-dependent Ca$^{2+}$ increase.

To gain further insight into the epistatic order of the Gq and Gs pathways, we overexpressed constitutive active forms of Gq pathway proteins in *Gαs* morphants with the *PC2 cis* element. If Gq is upstream of the Gs pathway, forced activation of the Gq pathway by caGαq or caPLCβ1/2/3 would not ameliorate the *Gαs* MO effect. Indeed, ca*PLCβ1/2/3*, which rescued *Gαq* morphants well, failed to rescue *Gαs* morphants (*Figure 4H–L*). In contrast, ca*Gαq* significantly ameliorated the metamorphosis-failed phenocopies of *Gαs* morphants (*Figure 4M*), contradicting the hypothesis that Gq is upstream of the Gs pathway. One possibility is that the Gq pathway stimulates cAMP synthesis through a massive Ca$^{2+}$ increase and protein kinase C activation (*Sadana and Dessauer, 2009*; *Cooper, 2015*). Indeed, *Ciona* larvae can synthesize cAMP without Gs, as suggested by the imaging of PF (*Figure 3—figure supplement 1C*), and by the partial rescue of *Gαs* morphants by theophylline treatment (*Figure 4N*). These data suggest that Gαs-independent AC could be a target of the Gq pathway. We concluded that the Gq pathway is upstream of the Gs pathway in the signaling cascade initiating *Ciona* metamorphosis.

## Interaction between GABA and Gq pathways

We next investigated the relationships between the GABA and Gq/Gs pathways. As our previous study showed, GABA pathway knockdown by *GAD* or *GABABR1* MO disrupted the initiation of metamorphosis (*Figure 5A*; *Hozumi et al., 2020*). Overexpression of ca*PLCβ1/2/3* and theophylline treatment ameliorated the metamorphosis-failed phenocopies of *GAD/GABABR1* morphants (*Figure 5B–D*). These results suggest that the GABA pathway is upstream of the Gq and Gs pathways. We observed Ca$^{2+}$ transients in the adhesive papillae of *GAD* knockdown larvae. *GAD* morphants rarely exhibited a Ca$^{2+}$ increase after stimulation (*Figure 5E*). Because the Ca$^{2+}$ transient in the papillae is Gq dependent, this result confirms that the GABA pathway is upstream of, or at least in parallel with, the Gq pathway.

However, puzzling results were obtained when we administered *Gαq*- and *Gαs*-knockdown larvae with GABA. If GABA is upstream of the Gq and Gs pathways, this chemical does not induce metamorphosis when the Gq or Gs pathway is disrupted. However, GABA partially but significantly ameliorated the metamorphosis-failed phenocopies of *Gαq*, *PLCβ*, and *Gαs* morphants (*Figure 5F–H*). Among the three morphants, GABA achieved more effective rescue in *Gas* knockdowns than *Gαq* or *PLCβ*. These results could be explained by assuming enhancement of the Gq pathway by GABA through PLCβ and another GABA-mediated metamorphic pathway bypassing Gq components. These possibilities are examined in the next section.

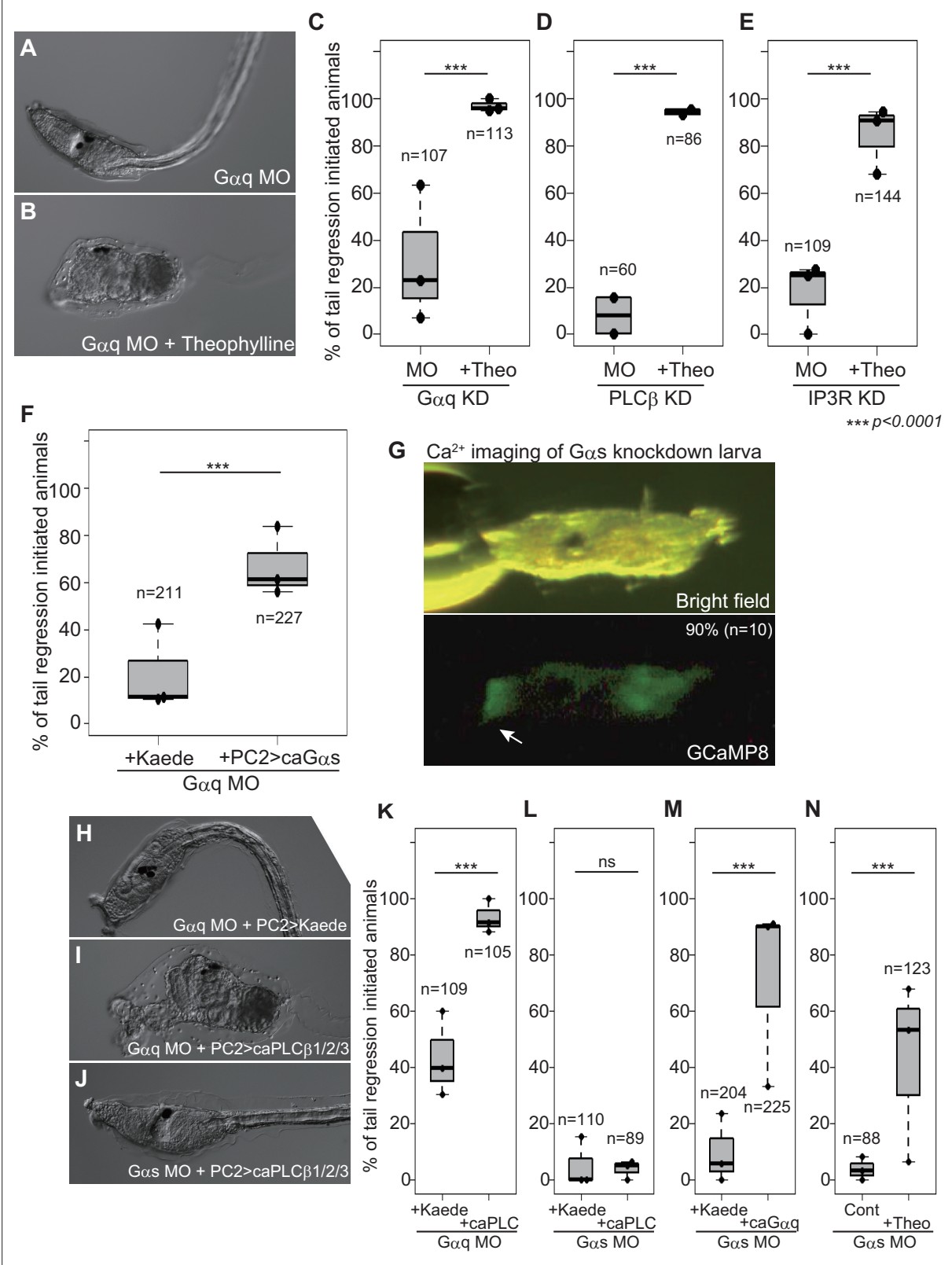

**Figure 4.** Relationships between Gq and Gs pathways in metamorphosis. (**A**) A *Gαq* knockdown larva. (**B**) Theophylline ameliorated the effect of *Gαq* knockdown. (**C**) Effect of theophylline on *Gαq* knockdown. ***, p<0.001 (Fisher's exact test). (**D**) Effect of theophylline on *PLCβ* knockdown. In this experiment, *PLCβ1/2/3* and *PLCβ4* were knocked down simultaneously. (**E**) Effect of theophylline on *IP3R* knockdown. (**F**) Effect of ca*Gαs* overexpression on *Gαq* knockdown. (**G**) *Gαs* is not necessary for $Ca^{2+}$ transient in the adhesive papillae (arrow) upon adhesion. The percentage and number

*Figure 4 continued on next page*

*Figure 4 continued*

represent the rate of animals showing Ca²⁺ transient in the papillae. (**H**) A *Gαq* knockdown and *Kaede*-overexpressed larva. (**I**) A *Gαq* knockdown and ca*PLCβ1/2/3*-overexpressed larva. (**J**) A *Gαs* knockdown and ca*PLCβ1/2/3*-overexpressed larva. (**K**) Effect of ca*PLCβ1/2/3* overexpression on *Gαq* knockdown. (**L**) Effect of ca*PLCβ1/2/3* overexpression on *Gαs* knockdown. (**M**) Effect of ca*Gαq* overexpression on *Gαs* knockdown. (**N**) Effect of theophylline treatment on *Gαs* knockdown.

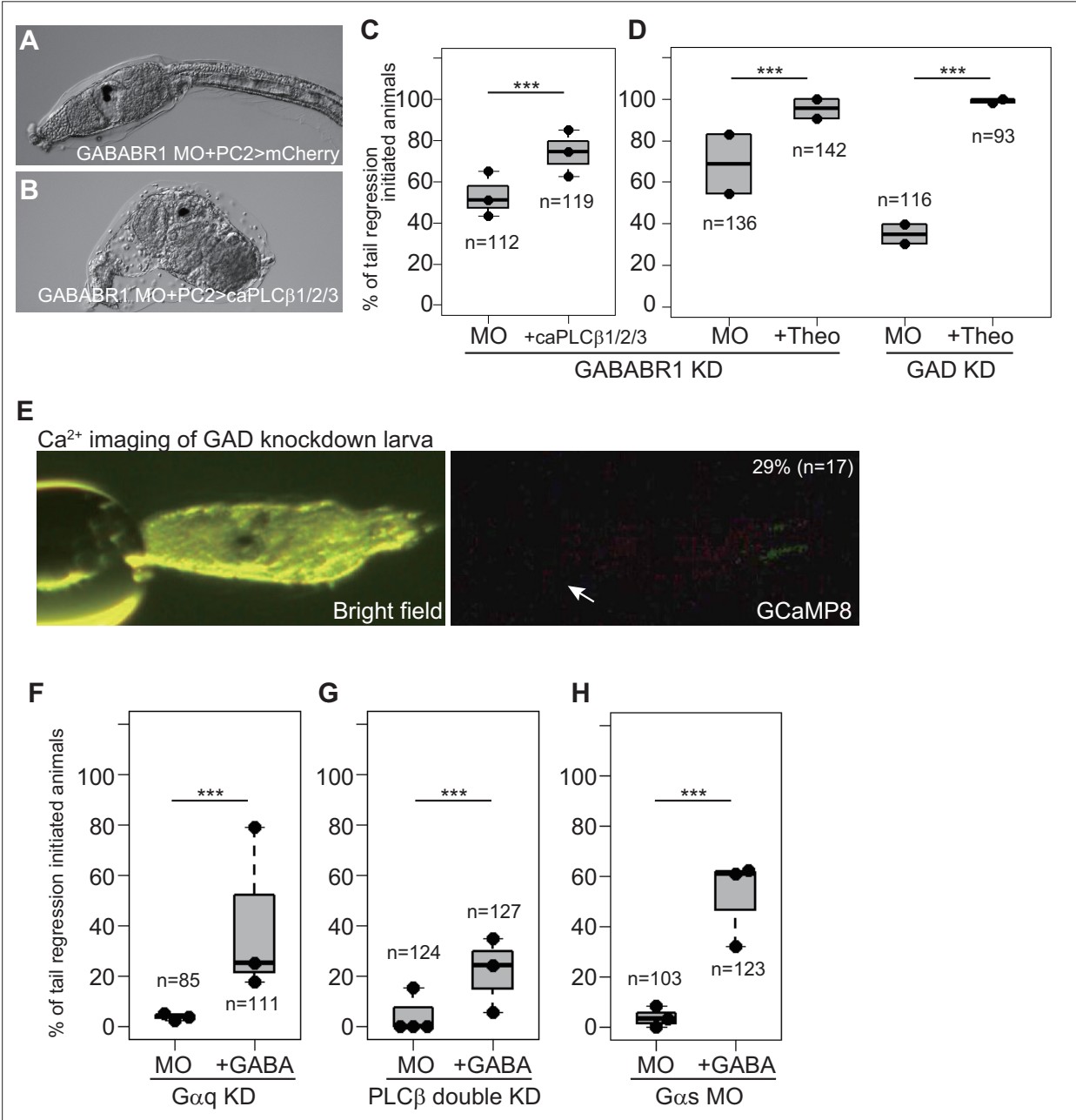

**Figure 5.** GABA functions in the adhesive papillae to induce metamorphosis. (**A**) A *GABABR1* knockdown and *mCherry*-overexpressed larva. (**B**) A *GABABR1* knockdown and ca*PLCβ1/2/3*-overexpressed larva. Metamorphosis initiated. (**C**) Effect of ca*PLCβ1/2/3* overexpression on *GABABR1* knockdown. ***, p<0.001 (Fisher's exact test). (**D**) Effect of theophylline on *GABABR1* and *GAD* knockdowns. (**E**) GABA is necessary for Ca²⁺ transient in the adhesive papillae (arrow). The percentage and number represent the rate of animals showing Ca²⁺ transient in the papillae. (**F**) Effect of GABA on *Gαq* knockdown. (**G**) Effect of GABA on *PLCβ* knockdown. (**H**) Effect of GABA on *Gαs* knockdown.

## Contribution of Gi to metamorphosis

The function of GABA as the potential upstream factor of Gq could be easily explained if GABABR is coupled with Gq. Although a few studies suggest this possibility (*Karls and Mynlieff, 2015*), Gi is regarded as the major G-protein coupled with GABABR (*Shen et al., 2021*). A previous study demonstrated that the gene encoding typical *Ciona* Gαi is strongly expressed throughout the entire larval nervous system, including in adhesive papillae (*Yoshida et al., 2004*), and our RNA-seq confirmed this (*Figure 1—figure supplement 1C* and *Supplementary file 1*). The knockdown of this gene (*Gαi*) exhibited a moderate (although statistically significant) reduction of metamorphosis rate (*Figure 6—figure supplement 1A*), suggesting the presence of another Gαi regulating metamorphosis. Indeed, the *dvGαi_Chr2* knockdown larvae failed to initiate metamorphosis (*Figure 6A–C*), and this phenocopy was rescued by theophylline and GABA (*Figure 6D*), suggesting that *dvGαi_Chr2* is chiefly necessary for metamorphosis. The knockdown of *dvGαi_Chr4* resulted in a slight reduction of the metamorphosis rate (*Figure 6—figure supplement 1B*), suggesting the supportive role of this Gαi in the metamorphosis initiation. These data strengthen the hypothesis that GABABR regulates metamorphosis through Gi. The rescue of *dvGαi_Chr2* morphants by GABA might be attributable to a redundant function of the typical Gαi and dvGαi_Chr4 proteins in the papillae; however, we could not fully address this hypothesis due to a technical limitation. That is, the simultaneous knockdown of *dvGαi_Chr2* and *dvGαi_Chr4* did not suppress the amelioration of metamorphosis by GABA (*Figure 6—figure supplement 1C*). This might have been due to the presence of typical *Gαi*. The simultaneous knockdown of *dvGαi_Chr2* and typical *Gai* resulted in malformation of the body (*Figure 6—figure supplement 1D*), and we could not examine their responsiveness to GABA.

We further explored the involvement of Gi upon metamorphosis as the upstream factor of the Gq–Gs pathways. Knockdown of *dvGαi_Chr2* resulted in the abolishment of temporal cAMP decrease and its subsequent accumulation upon adhesion (*Figure 6E*). Because these cAMP fluctuations are Gq and Gs dependent (*Figure 3—figure supplement 1C, D*), their dependence on dvGai_Chr2 supports that Gi is upstream of the Gq–Gs pathways. Gi is known to activate PLCβ through the Gβγi complex (*Tu et al., 2010*; *Mizuta et al., 2011*). Released βγ is inactivated by overexpressing normal (usually GDP-bound) Gα subunits because GDP-Gα quenches the βγ complex (*Sankaran et al., 1998*). Overexpressing wild-type *Gαi* and *Gαs* in the nervous system significantly suppressed metamorphosis (*Figure 6F*), suggesting that the activated βγ complex is necessary for initiating metamorphosis. To confirm that the wild-type Gαi exerts its effect through sequestration of the βγ complex, we overexpressed a dominant-negative form of Gαi (dnGαi) (*Barren and Artemyev, 2007*; *Slepak et al., 1995*) which has reduced GDP/GTP affinity while maintaining βγ-binding activity. dnGαi significantly reduced the occurrence of metamorphosis (*Figure 6—figure supplement 1E*), suggesting that the negative effect of Gαi on metamorphosis occurs through interaction with the βγ complex.

If Gβγ-mediated PLCβ activation is used in *Ciona* metamorphosis, PLCβ receives two independent inputs (Gαq and Gβγi) for its activation. These pathways could compensate for each other. We noticed that the MOs for GABA pathway genes and the *Gαq* MO did not disrupt metamorphosis as strongly as the *Gαs* MO (*Figures 1E, F and 5C, D*). The compensatory role could explain this phenomenon. Indeed, the simultaneous knockdown of *GABABR1* and *Gαq* resulted in a strong impairment of metamorphosis (*Figure 6G*), and these morphants initiated metamorphosis by overexpression of ca*PLCβ1/2/3* or ca*Gαs* (*Figure 6H*).

If the role of the GABA/Gi pathway relies specifically on PLCβ activation in the mechanism of metamorphosis, GABA administration would not rescue *PLCβ* morphants. However, GABA weakly but significantly induced metamorphosis in *PLCβ1/2/3* plus *PLCβ4* double-knockdown larvae (*Figure 5G*). Therefore, GABA/Gi is likely to activate another pathway that bypasses PLCβ. Gβγi is known to activate group III AC (*Sadana and Dessauer, 2009*). Its *Ciona* counterpart (AC5/6) is expressed in the adhesive papillae (*Figure 1—figure supplement 3B*), which could explain the results of the rescue experiments. In addition, there are two other major targets of Gβγi. One is the G-protein-activated inwardly rectifying potassium (GIRK) channel (*Dascal and Kahanovitch, 2015*). Our RNA-seq data on the papillae indicated the expression of two GIRK channel genes (*Supplementary file 2*). The GIRK channel negatively regulates the excitation of neurons through hyperpolarization. If the metamorphosis of *Ciona* is induced by the excitation of PNs as suggested previously (*Wakai et al., 2021*; *Sakamoto et al., 2022*), the GIRK channel is likely to regulate metamorphosis negatively. Indeed, the knockdown of one GIRK channel gene weakly enhanced the initiation of metamorphosis without

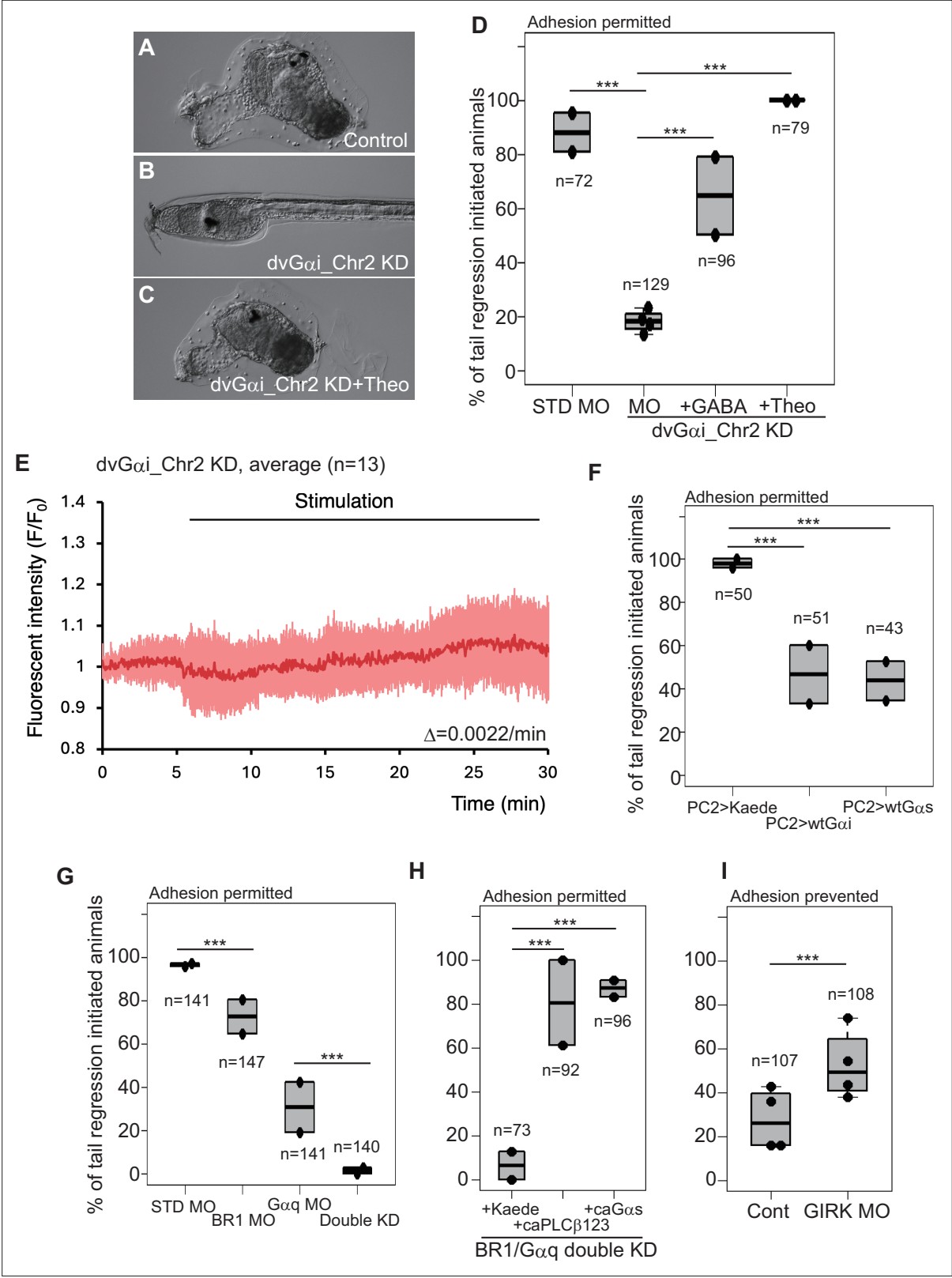

**Figure 6.** Gi is required for metamorphosis initiation. (**A**) A control animal injected with STD MO. (**B**) A *dvGαi_Chr2* knockdown larva. (**C**) A *dvGαi_Chr2* knockdown and theophylline-treated animal. (**D**) Effects of GABA and theophylline on *dvGαi_Chr2* knockdown. ***, p<0.001 (Fisher's exact test). (**E**) Measurement of cAMP concentration in the *dvGαi_Chr2* knockdown larvae. The red and pink graphs represent the averaged Pink Flamindo fluorescent intensity and the standard deviations, respectively. The number in the parentheses indicates the number of examined larvae. The red graph

*Figure 6 continued on next page*

*Figure 6 continued*

is approximated to the linear graph to calculate its slope, which is shown as Δ. See also Figure S4. (**F**) Effects of wt*Gαi* and wt*Gαs* overexpression. (**G**) Effect of *GABABR1* and *Gαq* simultaneous knockdown. (**H**) Effects of ca*PLCβ1/2/3* and ca*Gαs* overexpressions on *GABABR1* and *Gαq* simultaneous knockdown. (**I**) *GIRK* knockdown promoted metamorphosis initiation.

The online version of this article includes the following figure supplement(s) for figure 6:

**Figure supplement 1.** Gi is required for metamorphosis.

**Figure supplement 2.** Autonomous activity of wild-type Gαq.

settlement (*Figure 6I*). Although the negative role of the GIRK channel supports the involvement of Gβγi in the pathway of metamorphosis, this does not explain the PLCβ-independent activation of the metamorphic pathway by GABA/Gi.

The third function of Gβγi is to activate MAPK signaling through positive regulation of MEK (*Blaukat et al., 2000*). PLCβ does not mediate this pathway. It has been suggested that MAPK signaling is required to induce metamorphosis (*Chambon et al., 2002*; *Chambon et al., 2007*). To show that MEK activation is necessary for inducing metamorphosis mediated by GABA, we treated adhesion-prevented larvae with GABA plus U0126, a potent MEK1/2 inhibitor repeatedly used in ascidians (*Kim and Nishida, 2001*; *Hudson et al., 2003*). U0126 significantly reduced the rate of metamorphosis induction by GABA (*Figure 6—figure supplement 1F*).

cAMP is also known to activate the pathway that involves MEK1/2 (*Goldsmith and Dhanasekaran, 2007*). We found that U0126 antagonized the effect of theophylline on metamorphosis (*Figure 6—figure supplement 1G*). Therefore, GABA and cAMP have the same target (MEK1/2) to initiate metamorphosis, suggesting their compensatory function and/or that cAMP synthesis is upregulated by GABA-Gβγi. These bypassing pathways could explain the amelioration of the phenocopies of *PLCβ* and *Gαs* morphants by GABA (*Figure 5G, H*).

## The constitutive function of wild-type Gαq

We found that *Ciona* wild-type Gαq (wtGαq) has constitutive activity. In contrast to wt*Gαs* and wt*Gαi*, overexpression of wt*Gαq* did not arrest metamorphosis (*Figure 6—figure supplement 2A*). Rather, its overexpression induced metamorphosis without settlement (*Figure 6—figure supplement 2B*). The constitutive wtGαq activity was confirmed by the significant rescue of *Gαs* morphants (*Figure 6—figure supplement 2C*). As mentioned above, caGαq also has these activities (*Figure 1L*). caGαq-overexpressed larvae had a somewhat rounder trunk shape, abnormal tail bending, and frequent tail twitching, perhaps due to a constitutive increase in the cytosolic $Ca^{2+}$ concentration (*Figure 6—figure supplement 2D*). Overexpression of wt*Gαq* did not show such abnormalities (*Figure 6—figure supplement 2E*), suggesting that wtGαq has milder activity than, or works through a different mechanism from, caGαq. To gain further insight into the constitutive effect of wtGαq, we constructed a dominant-negative form of Gαq (*Barren and Artemyev, 2007*) that has reduced GDP/GTP-binding activity while maintaining βγ-binding activity. Overexpression of dn*Gαq* weakly inhibited metamorphosis (*Figure 6—figure supplement 2F*), suggesting that GDP/GTP exchange is important for the constitutive activity of wtGαq.

## Discussion

Among chordates, the tunicate ascidian is the only group that exhibits a sessile lifestyle at the adult stage. Understanding how ascidians acquired the mechanism to metamorphose into sessile adults is important for elucidating the evolution of chordates (*Ferrández-Roldán et al., 2021*; *Nanglu et al., 2023*). Through extensive molecular, physiological, and pharmacological analyses, we identified signaling molecules responsible for the initiation of metamorphosis of the ascidian *Ciona*. Combining the results with previous knowledge, we described a schematic that best represents the cascades running in the adhesive papillae upon adhesion to initiate metamorphosis (*Figure 7*). This working hypothesis will be the basis for future research on ascidian/tunicate metamorphosis. The hypothesis explains the characteristics of *Ciona* metamorphosis, the mechanisms of which have not yet been elucidated. The genes, proteins, and signaling pathways in this schematic are essential targets for

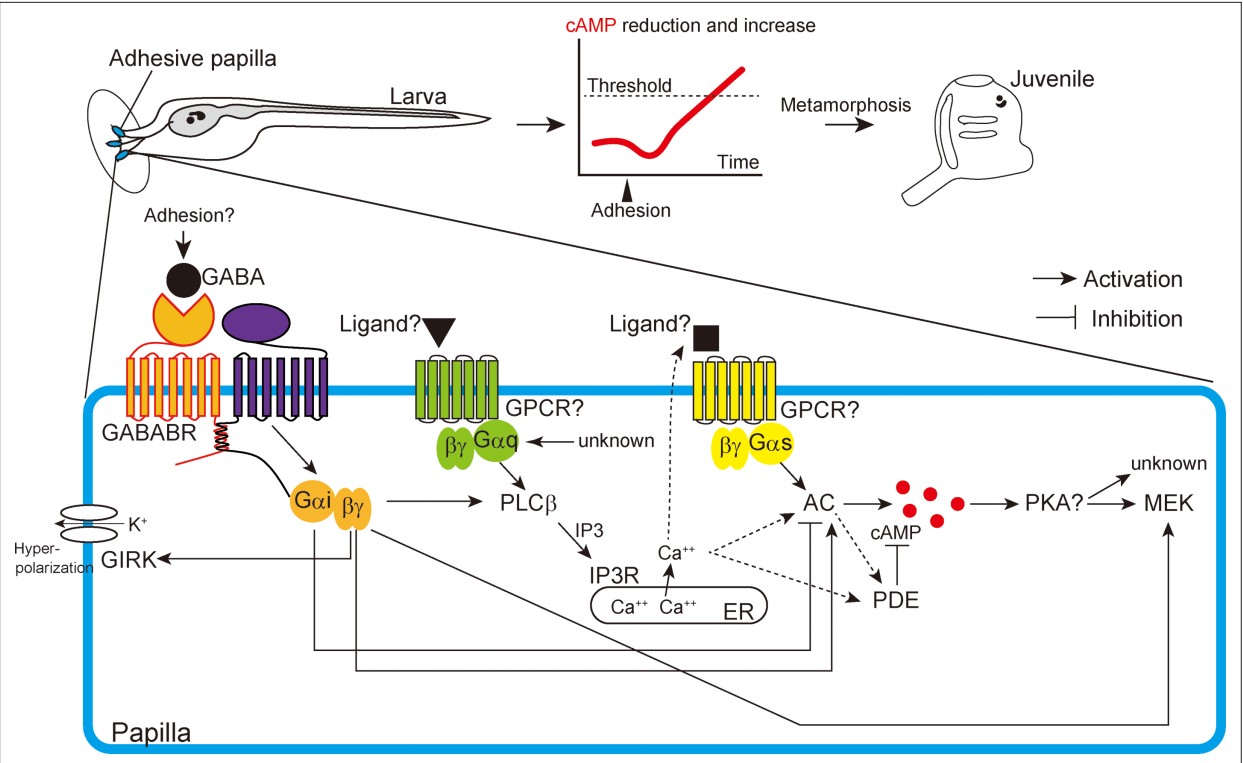

**Figure 7.** Schematic illustration of the signaling cascades for initiating *Ciona* metamorphosis. AC, adenylate cyclase; ER, endoplasmic reticulum; GPCR, G-protein-coupled receptor; PKA, protein kinase A. Figure legends for figure supplements and Supplementary files.

elucidating how the ancestor of ascidians acquired the metamorphosis system peculiar to this group during evolution.

## Mechanisms measuring duration and strength of adhesions

One mystery of *Ciona* metamorphosis is that its initiation requires continuous adhesion with the adhesive organ for a certain period. This requirement is suggested for the faithful achievement of metamorphosis only when adhesion is firm enough to allow the transition out of the free-living larval stage. Our previous study suggested that approximately 30 min of adhesion is necessary (*Matsunobu and Sasakura, 2015*). A shorter duration of adhesion does not trigger metamorphosis. Larvae that experience short-term adhesion (such as detaching before 30 min) need to adhere again for 30 min to initiate metamorphosis, suggesting that the experience of temporal adhesion is erased. Our recent study showed that the force given to the papillae by adhesion also affects metamorphosis initiation: weaker force extends the time requirement (*Sakamoto et al., 2022*). Therefore, *Ciona* larvae likely possess a mechanism that somehow measures the duration and strength of adhesion and the ability to cancel transient excitations of the adhesive organ. Because fewer than 300 neurons constitute the larval nervous system of *Ciona* (*Ryan et al., 2016*; *Ryan et al., 2018*; *Ryan and Meinertzhagen, 2019*), how the larva measures the strength and duration of adhesion with its simple nervous system is an important question to address the mechanisms of metamorphosis.

This study showed that cAMP is an essential second messenger molecule for triggering metamorphosis. Because PDE constitutively degrades cAMP, accumulation of this molecule requires persistent and/or strong activation of the metamorphic pathway that overcomes PDE activity. In other words, the duration and strength of adhesion could be converted into the quantity of cAMP, and when its quantity reaches a threshold, metamorphosis can be initiated. The time between the start of adhesion and metamorphosis initiation may be the period necessary to accumulate sufficient cAMP. This 'cAMP timer' mechanism does not demand a complicated neuronal network system and could work in the simple nervous system of *Ciona*. Prior to triggering the cAMP timer, *Ciona* temporarily decreases cAMP quantity (*Figure 3E*, *Figure 3—figure supplement 1A*). This reduction is also associated with

metamorphosis initiation, suggesting that *Ciona* does not simply sense the cAMP increase. There are two plausible functions of this reduction. One is guaranteeing that the requirement of long-term adhesion will be met until a sufficient quantity of cAMP is accumulated, irrespective of the initial quantity of cAMP. The other one is multiplying the degree of cAMP increase. By reducing the initial value, larvae can increase the amount of this molecule several fold, even though the maximum quantity remains the same. Such an elevated difference in cAMP quantity may be adopted to ensure that the timing of metamorphosis initiation is appropriate using this ubiquitous molecule.

## G-protein signaling relay for metamorphosis

This study showed that three major G-proteins are involved in the initiation of metamorphosis (*Figure 7*). This complicated mechanism stands in contrast to the simple determination of metamorphosis initiation by cAMP quantity. What factor promoted the ancestor of ascidians to acquire this complicated signaling network of metamorphosis? We suspect that incorporating multiple components increases the chance of generating the crosstalk between molecules, providing robustness and flexibility to the system. Particularly, the involvement of Gi may have been important for achieving metamorphosis at the correct time and condition. GABA and Gi have inhibitory and excitatory functions (*Ben-Ari et al., 2007*; *Pfeil et al., 2020*). *Ciona* larvae must ignore transient adhesion or stimulation on the papilla, which may occur frequently during swimming by colliding with an object or by strong water flows. At the same time, larvae must maintain sensing ability and start metamorphosis when they encounter an appropriate stimulus. Through the GIRK channel and Gαi, GABA can suppress the excitation and cAMP accumulation in the papilla. The GIRK channel could increase the excitation threshold of PNs, allowing them to ignore weak stimuli. The transient reduction of cAMP at the onset of adhesion could be partly explained by the inhibitory function of Gαi on ACs. Not only Gi, but Gq and Gs are also necessary for this cAMP reduction. PDE1 is known to be activated by $Ca^{2+}$/calmodulin (*Azevedo et al., 2014*). Because PDE1 is abundantly expressed in the papillae (*Figure 1—figure supplement 3B*), the increase in $Ca^{2+}$ upon adhesion by Gi and Gq could also trigger sudden cAMP reduction through PDE1. The contribution of Gs to cAMP reduction is surprising because the general role of this complex is to increase cAMP. However, it is known that an important target of the Gs pathway for the termination of cAMP synthesis is PDE (*Azevedo et al., 2014*). The involvement of three G-protein pathways suggests that the acute cAMP reduction is a special event that is distinct from its constitutive degradation.

Together with its inhibitory function, GABA can activate the Gq pathway, which promotes metamorphosis through Gβγi. Group III AC and MEK could also be the targets of Gβγi. When adhesion is long enough, their activations somehow overcome the inhibitory functions of GABA. The GABA pathway is an ideal player that fulfills multiple requirements in the regulation of metamorphosis through its excitatory and inhibitory functions. We suspect that papillae themselves are the source of GABA for metamorphosis initiation. Because VIAAT/VGAT is not expressed in the papillae, the secretion of GABA from the PNs requires an atypical mechanism. GABA is detected in the cell body of the neurons (*Brown et al., 2005*). The release of cytosolic GABA may be triggered by the reception of mechanical stimuli using the TRP channel (*Sakamoto et al., 2022*). A VIAAT-independent cytosolic GABA release is observed in pancreatic beta cells (*Menegaz et al., 2019*). Curiously, PNs express the transcription factor Islet (*Wagner et al., 2014*), which is homologous to the vertebrate transcription factor responsible for β cell function. The atypical secretion of GABA might depend on a transcription factor like Islet shared between *Ciona* PNs and vertebrate beta cells.

A recent study (*Hoyer et al., 2024*) provided several lines of evidence suggesting that PNs can serve as the sensors of several chemicals in addition to mechanical stimuli. This finding and our model could be mutually related because these chemicals could modify $Ca^{2+}$ and cAMP production. G-protein signaling allows *Ciona* to reflect various environmental stimuli to initiate metamorphosis either mechanically or chemically according to the situation.

## GPCRs for initiating metamorphosis and atypical Gαq activity

Future studies need to specify the mechanisms underlying Gαi, Gαq, and Gαs activation as well as the interaction between them more precisely. The activation of trimeric G-proteins relies on GPCRs, suggesting the presence of the receptors coupled with Gi, Gq, and Gs for metamorphosis initiation (*Figure 7*). Among them, Gi is likely to be coupled with GABABR; however, the expression of

GABABR in the papillae is weak, and their interaction needs to be clarified in the future. We found that wild-type Gαq exhibits constitutive activity that does not require activation through adhesion. The constitutive Gαq activity requires GTP binding. If a GPCR owns this Gαq activation, its ligand should be supplied constitutively. Wild-type *Ciona* does not initiate metamorphosis without adhesion, even though Gαq is expressed in the adhesive papillae, suggesting that the quantity of Gαq is strictly regulated to prevent autonomous metamorphosis in normal conditions.

We did not identify the GPCR coupled with Gs in the metamorphic mechanism. We suspect that GnRH receptors (GnRHRs) are strong candidates for this role. Our previous studies showed that GnRHs can induce metamorphosis (*Hozumi et al., 2020*; *Kamiya et al., 2014*) as a downstream factor of GABA. Among the four genes encoding GnRHR proteins, *GnRHR1* and *GnRHR2* are expressed in the adhesive papillae (*Kusakabe et al., 2012*). These GnRHRs stimulate cAMP signaling, and the ligand GnRHs for GnRHR1 are also expressed in the papillae (*Kusakabe et al., 2012*; *Tello et al., 2005*; *Sakai et al., 2020*). Therefore, GnRHs could be secreted from adhesive papillae as downstream molecules of GABA and stimulate the Gs pathway through GnRHR1. Future studies targeting the regulation of the secretion and reception of GnRH peptides after stimulation of papillae will improve our understanding of the signaling cascade conducting *Ciona* metamorphosis.

Understanding whether the G-protein signaling relay occurs in cell- or non-autonomous fashions is crucial for deepening our understanding of *Ciona* metamorphosis. Each papilla comprises approximately 20 cells, including four PNs. We showed that activation of Gq and Gs pathways in PNs is sufficient for initiating metamorphosis. This coincides with the previous reports (*Sakamoto et al., 2022*; *Johnson et al., 2023*) that the loss of PNs by disrupting the POUIV transcription factor caused the complete arrest of metamorphosis. However, we do not think PNs are the only cells that activate G-proteins, because $Ca^{2+}$ and cAMP imaging showed upregulation of fluorescence in the entire papillae (*Wakai et al., 2021*; *Sakamoto et al., 2022*). A recent study also reports $Ca^{2+}$ transient in another cell type than PNs (*Hoyer et al., 2024*). Other papilla cells are likely responsible for activating G-proteins by directly responding to adhesion and by receiving signal input from adjacent cells, thereby enhancing the signal strength to accumulate a sufficient quantity of cAMP in the PNs to initiate metamorphosis. In this study, technical limitations prevented us from characterizing cells exhibiting increases in $Ca^{2+}$ and cAMP; we need to address this question in future studies.

## Evolutionary implications

Sessile or benthic marine invertebrates lose locomotive activity when they undergo metamorphosis triggered by adhesion to a substratum (*Jackson et al., 2002*). These animals are suspected of having a system to control the adhesion state, allowing them to repeat attachment and detachment before meeting the appropriate conditions for metamorphosis. For example, barnacle larvae 'walk' on the substratum before adhering firmly by secreting cement (*Lagersson and Høeg, 2002*). Like *Ciona*, the larvae of some sessile/benthic animals may have a system to initiate metamorphosis only when appropriate adhesion is provided, while erasing stimuli from transient and inappropriate adhesion. GABA serves as the metamorphosis inducer of some benthic invertebrates, including mollusks and echinoderms (*Morse et al., 1980*; *Rahmani and Ueharai, 2001*; *Pearce and Scheibling, 1990*). Moreover, GPCRs are implicated as the mediators of settlement and metamorphosis induction in hydrozoans, mollusks, and barnacles (*Morse et al., 1980*; *Leitz and Müller, 1987*; *Rittschof et al., 1986*). In the abalone and barnacle, the dual use of Gq and Gs pathways in metamorphosis has been suggested (*Clare, 1996*; *Holm et al., 1998*). Supported by these shared features in the metamorphic mechanisms, our working hypothesis about the initiation of *Ciona* metamorphosis (*Figure 7*) will serve as a cue to elucidate how marine benthic invertebrates regulate their metamorphosis.

## Materials and methods
### Animals

*C. intestinalis* Type A wild types collected from Onagawa Bay (Miyagi, Japan) and Onahama Bay (Fukushima, Japan) were cultivated in closed colonies by the staff of the National BioResource Project, Japan. They were kept under a constant light condition to prevent gamete release. Eggs and sperm were collected surgically from gonadal ducts, and insemination was carried out in dishes. To prevent larvae from initiating metamorphosis, the tail's posterior half was manually cut with a scalpel. Removing

the tail prevents larvae from swimming efficiently, and these larvae are usually unable to adhere to a substrate. These adhesion-prevented larvae do not metamorphose, since adhesion is required to initiate metamorphosis. Larvae developed from dechorionated eggs were cultured on a 2% agar-coated dish after tail amputation to prevent metamorphosis. Because these larvae stick to plastics, tail amputation is insufficient to avoid their adhesion to the culture dish.

## Pharmacological treatment

Larvae or larval anterior fragments isolated with a scalpel were administered overnight with 700 µM GABA (Fujifilm Wako #010-02441) or 1 mM theophylline (Sigma-Aldrich #T1633) dissolved in seawater, or 4 µM U0126 (Promega #V1121) dissolved in DMSO.

## Plasmids

The open reading frame (ORF) of Kaede was removed from pSPCiPC2Kaede by inverse PCR with PrimeStar GxL DNA polymerase (Takara-bio #R050). The ORFs of *Gαq*, *Gαs*, *Gαi*, and *PLCβ1/2/3* were PCR amplified using full-insert cDNAs (*Roure et al., 2007*). These PCR fragments were fused with the In-Fusion HD Cloning kit (Clontech #639650). The region encoding the XY linker (*Charpentier et al., 2014*) was deleted from *PLCβ1/2/3* cDNA by inverse PCR to create a constitutively active form. The mutations were introduced by inverse PCR to create constitutively active or dominant-negative forms of *Gα* cDNAs. The introduced mutations are as follows: ca*Gαq*, corresponding to Q223L, abolishes GTPase activity (*Oki et al., 2005*); ca*Gαs*, corresponding to Q227L, abolishes GTPase activity (*Kelly et al., 2007*); dn*Gαi*, corresponding to S47C (*Barren and Artemyev, 2007*; *Slepak et al., 1995*); and dn*Gαq*, corresponding to S54N (*Barren and Artemyev, 2007*). The ORF of *bPAC* (*Stierl et al., 2011*) was PCR amplified and inserted between the *Bam*HI and *Eco*RI restriction sites of pSP-Kaede (*Hozumi et al., 2010*) to create pSPbPAC. The PCR-amplified *PC2 cis*-element (*Osugi et al., 2017*) was inserted into the *Bam*HI site of pSPbPAC. The plasmid DNAs were linearized by a restriction enzyme and purified with the Qiaquick Gel Extraction Kit (QIAGEN #28706) before microinjection. The ORF of *Pink Flamindo* (*Harada et al., 2017*) was PCR amplified and inserted into the *Eco*RV restriction sites of pBS-HTB (*Akanuma et al., 2002*; *Sasakura et al., 2010*) to create pHTBPinkFlamindo. The plasmids pHTBGCaMP8 (*Sakamoto et al., 2022*) and pHTBPinkFlamindo were linearized using *Xho*I for subsequent in vitro synthesis of *GCaMP8* and *Pink Flamindo* mRNA, respectively. *GCaMP8* mRNA was synthesized with the MEGAscript T3 kit (Thermo Fisher Scientific #AM1338), the Poly (A) Tailing kit (Thermo Fisher Scientific #AM1350), and the Cap structure analog (New England Biolabs #S1404). PF mRNA was synthesized with the mMESSAGE mMACHINE T3 Transcription kit (Thermo Fisher Scientific #AM1348). The plasmids used in this study are available through the National BioResource Project (NBRP) Japan.

## Microinjection

Unfertilized eggs were dechorionated in sterilized seawater containing 1% sodium thioglycolate (Fujifilm Wako #590-11762) and 0.05% actinase E (Kaken Pharmaceutical #650164) as previously described (*Kobayashi and Satou, 2018*). The microinjection solution included 2 mg/ml of Fast Green FCF (Fujifilm Wako #061-00031), 0.5–1.0 mM of MOs (*Satou et al., 2001*), 5 ng/µl of plasmids, and/or 1 mg/ml of *GCaMP8* mRNA for imaging (*Sakamoto et al., 2022*). For cAMP imaging, 1 mg/ml *Pink Flamindo* mRNA was dissolved in water without Fast Green, since this chemical emits fluorescence. Microinjected unfertilized eggs were inseminated in a gelatin-coated plastic dish. After the seawater was exchanged to remove excess sperm, the fertilized eggs were cultured at 18 or 20°C overnight until the larval stage. MOs are listed below. Standard control MO, 5'-CCTCTTACCTCAGTTACAATTTATA-3'; *GAD* ATG MO (*Hozumi et al., 2020*), 5'-ACCTCCAAGCCGATTGTTTCTGCAT-3'; *GABABR1* ATG MO (*Hozumi et al., 2020*), 5'-GCTTACGACTTTACATAACCTTACA-3'; *Gαq* ATG MO, 5'-GGCATATTTGTG ACTATAATGACG-3'; *Gαs* ATG MO, 5'-AAAGCAACCCATTGGCATTATCGAC-3'; *PLCβ1/2/3* splicing MO, 5'-GTGTTACTTACGCTTTCTCTA-3'; *PLCβ4* splicing MO, 5'-AACCACCAACCACCAACCTTTTG-3'; *IP3R* splicing MO, 5'-AATGATGGTTTAAAATTGCCACCTG-3'; *Gαi* ATG MO, 5'-GTGGAGACTGTG CAACCCATGATTC-3'; *dvGai_Chr2* ATG MO, 5'-CCATCTTGAGTAATCCAGGCTTTTA-3'; *dvGai_Chr4* ATG MO, 5'-CATGGTCAGCGGTTTACAAAGTATT-3'; *GIRK channel* ATG MO, 5'-TCTGCTGGTTCA GTAATAGACATAG-3'.

## Photographing and imaging

Photographs were taken with an AxioImager Z1 and AxioObserver Z1 (Carl Zeiss). The images were treated with AxioVision Rel.4.6 or Zen (Carl Zeiss) and Photoshop 2021 (Adobe) software. Imaging of the $Ca^{2+}$ transient was carried out according to previous reports (*Wakai et al., 2021*; *Sakamoto et al., 2022*). The tails of *GCaMP8* mRNA-injected larvae were removed by a scalpel at 15–17 hpf. At 30–40 hpf, the larvae were mounted onto a plastic dish under an M165FC stereo microscope (Leica Microsystems), and their adhesive papillae were stimulated by a glass rod using a three-axis coarse manipulator M-152 (Narishige Japan). Immediately after the tip of the rod had contact with the papillae, visible light was shut off and the fluorescence of GCaMP8 was taken with a DFC 310FX digital camera (Leica), Las version 4.12 (Leica), and the movie-capturing function of Windows 10. Images were treated with Photoshop 2021.

Imaging of cAMP was taken by confocal laser scanning microscopy (FV1000, Olympus). The movies were treated with ImageJ (National Institutes of Health). The PF mRNA-injected larvae were immobilized on poly-L-lysine-coated glass-bottom dishes at 20–21 hpf, and their adhesive papillae were stimulated around 25 hpf. For the stimulation, an electric manipulator MM-89 (Narishige) and hydraulic micromanipulator MMO-202ND (Narishige) were used.

## RNA sequencing

Larval anterior portions, including adhesive papillae and the other trunk region, were separately isolated by a scalpel around 22–25 hpf at 18°C and collected in Isogen (Nippon Gene #317-02503) soon after isolation. To identify GABA-responsive genes, half of the tail was amputated from larvae around 21–24 hpf, followed by GABA administration for 2 hr before treatment with Isogen. In each experiment, approximately 50 fragments were collected, and two biological replicates were taken. Total RNA was isolated according to the manufacturer's instructions. Ribosomal RNA was depleted from the total RNA using the NEBNext rRNA Depletion kit (New England Biolabs #E6310), followed by the conversion of the remaining RNA into an Illumina sequencing library using the NEBNext Ultra Directional RNA Library Prep kit (New England Biolabs #E7760). Following library preparation, validation was performed using the Bioanalyzer system (Agilent Technologies) to assess size distribution and concentration. Subsequently, sequencing was carried out on the NextSeq 500 platform (Illumina) employing the paired-end 36-base read option. Reads were mapped on the *C. intestinalis* Type A reference genome (HT model; http://ghost.zool.kyoto-u.ac.jp/default_ht.html) and quantified using CLC Genomic Workbench version 22.0 (QIAGEN). RNA-seq data sets were deposited in the NCBI Gene Expression Omnibus (GEO) under accession numbers SAMN40712866 to SAMN40712873, which will appear online upon publication of this paper.

The read counts were normalized by calculating the number of reads per kilobase per million for each transcript in each sample using CLC Genomic Workbench version 22.0 (QIAGEN). Eight candidate genes showing a minimum 1.5-fold increase following GABA administration, along with notably higher expression in the papillae in comparison to the trunk (an approximately 4-fold increase in a sample), were chosen for quantitative PCR analysis to assess their response to GABA.

## Quantitative PCR

Total RNA was isolated from the anterior tips of larvae treated with the chemical using Isogen (Nippon Gene), following the manufacturer's instructions. After isopropanol extraction, genomic DNA removal and reverse transcription were carried out using the PrimeScript RT reagent kit with gDNA Eraser (Takara Bio #RR047). Quantitative RT-PCR (qRT-PCR) was done using a SYBR Premix Ex TaqII Dimer Eraser (Takara Bio #RR091) and a Thermal Cycler Dice Real Time System III (Takara Bio). The gene encoding elongation factor 1α (*EF1α*) was used to normalize RNA quantity according to a previous study (*Sasakura et al., 2010*). qPCR primers are listed below. KY21.Chr5.240, 5′-TCTTCTCAAAGT TGCACATTCC-3′ and 5′-CAGCAGCAACCAAACGATAAAC-3′; KY21.Chr8.489, 5′-CAATGCAACTTT GACTGCATAC-3′ and 5′-TCCAAACTGCATTCCACATATC-3′; *EF1α*, 5′-CATGTCACGGACAGCGAAAC G-3′ and 5′-CAATGTGTGTTGAGGCATTCCAAG-3′.

## Statistical analysis

Differences between conditions were evaluated by Fisher's exact test. Statistical analyses and most graph visualizations were conducted utilizing R x64.4.1.2 and RStudio software.

## Phylogenetic analysis

We performed a tBlastn search (*Altschul et al., 1997*) against the HT gene model of *Ciona* (*Satou et al., 2022*) using the amino acid sequences of human PDE, PLCβ, and *Ciona* Gα proteins as queries, followed by a reciprocal Blast search. We aligned the amino acid sequences from the HDc domain of PDE, the EF-hand_like, PLCXc, PLCYc, and C2 domains of PLCβ, and the whole ORF of Gα proteins using the M-COFFEE program (*Notredame et al., 2000*; *Wallace et al., 2006*). After the removal of unnecessary residues, maximum likelihood trees were constructed using MEGA software version 11 (*Tamura et al., 2021*), employing the WAG amino acid substitution matrix (*Whelan and Goldman, 2001*). The trees were assessed with 1000 bootstrap pseudoreplicates.

## Acknowledgements

We would like to express our earnest gratitude to Drs. Kazuki Horikawa, Atsuo Nishino, Ryusuke Niwa, and Takahiro Yamashita for their kind material provision, helpful comments, and general support of our study. We thank Dr. Kogiku Shiba for instructing us with the pharmacological analyses. We also thank Dr. Masafumi Muratani and members of i-Laboratory at the University of Tsukuba for their support for RNA sequencing. We are grateful to the past and present members of the Shimoda Marine Research Center at the University of Tsukuba for their contributions to the initial step of this study and the maintenance of the animals. We thank Drs. Shigeki Fujiwara, Manabu Yoshida, Yutaka Satou, and all members of the Department of Zoology, Kyoto University, the Misaki Marine Biological Station, the University of Tokyo, the Maizuru Fishery Research Station of Kyoto University, and the National BioResource Project (NBRP) for the cultivation and provision of *Ciona* adults and experimental materials. This study was supported by grants from the Japan Society for the Promotion of Science to K.H. (21H00440, 23H04717, 24K02038), T.H. (19H03204, 21K19249, 21H05239), and Y.S. (19H03262). Y.S. was further supported by a Takeda Bioscience Research Grant. T.H. was supported by the Collaborative Research in Computational Neuroscience Program (CRCNS2021). N.M.T was supported by JST SPRING (JPMJSP2123).

## Additional information

### Funding

| Funder | Grant reference number | Author |
|---|---|---|
| Japan Society for the Promotion of Science | 21H00440 | Kohji Hotta |
| Japan Society for the Promotion of Science | 23H04717 | Kohji Hotta |
| Japan Society for the Promotion of Science | 24K02038 | Kohji Hotta |
| Japan Society for the Promotion of Science | 19H03204 | Takeo Horie |
| Japan Society for the Promotion of Science | 21K19249 | Takeo Horie |
| Japan Society for the Promotion of Science | 21H05239 | Takeo Horie |
| Japan Society for the Promotion of Science | 19H03262 | Yasunori Sasakura |
| Takeda Bioscience Research Grant | 2023 | Yasunori Sasakura |
| Collaborative Research in Computational Neuroscience Program | CRCNS2021 | Takeo Horie |
| Japan Science and Technology Agency | JPMJSP2123 | Nozomu M Totsuka |

| Funder | Grant reference number | Author |
|---|---|---|

The funders had no role in study design, data collection, and interpretation, or the decision to submit the work for publication.

## Author contributions

Akiko Hozumi, Conceptualization, Resources, Data curation, Formal analysis, Validation, Investigation, Visualization, Methodology, Writing – review and editing; Nozomu M Totsuka, Resources, Data curation, Formal analysis, Validation, Investigation, Visualization, Methodology, Writing – review and editing; Arata Onodera, Yanbin Wang, Data curation, Formal analysis, Validation, Investigation, Visualization; Mayuko Hamada, Akira Shiraishi, Data curation, Formal analysis, Validation; Honoo Satake, Supervision; Takeo Horie, Data curation, Formal analysis, Supervision, Funding acquisition; Kohji Hotta, Supervision, Funding acquisition, Writing – review and editing; Yasunori Sasakura, Conceptualization, Resources, Data curation, Formal analysis, Supervision, Funding acquisition, Validation, Investigation, Visualization, Methodology, Writing – original draft, Project administration, Writing – review and editing

## Author ORCIDs

Mayuko Hamada  https://orcid.org/0000-0001-7306-2032
Akira Shiraishi  https://orcid.org/0000-0003-3456-3074
Honoo Satake  https://orcid.org/0000-0003-1165-3624
Kohji Hotta  https://orcid.org/0000-0003-4614-7473
Yasunori Sasakura  https://orcid.org/0000-0001-7161-8367

Reviewer #1 (Public review): https://doi.org/10.7554/eLife.99825.3.sa1
Reviewer #2 (Public review): https://doi.org/10.7554/eLife.99825.3.sa2
Author response https://doi.org/10.7554/eLife.99825.3.sa3

# Additional files

## Supplementary files

Supplementary file 1. Expression levels of genes related to G-protein signaling during metamorphosis, as revealed by RNA-seq of the larval anterior tip including papillae.

Supplementary file 2. List of candidate genes upregulated by GABA.

MDAR checklist

## Data availability

RNA-seq data sets were deposited in the NCBI BioProject, under accession number PRJNA1095187.

The following dataset was generated:

| Author(s) | Year | Dataset title | Dataset URL | Database and Identifier |
|---|---|---|---|---|
| Sasakura Y | 2024 | Comparison of gene expression between papilla-containing region and remaining trunk, and GABA- and solvent-treated larval trunk, as revealed by RNA-seq | https://www.ncbi.nlm.nih.gov/bioproject/PRJNA1095187 | NCBI BioProject, PRJNA1095187 |

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
