## [Editor Report · eLife Assessment]

This **important** work substantially advances our understanding of the molecular mechanisms underlying the timing of the initiation of metamorphosis of the Ciona ascidian tadpole larva. Through the combination of gene knockdown experiments and fluorescent molecular reporters the authors provide **compelling** evidence about a crosstalk between different G protein mediated signalling pathways and are able to place different signalling molecules within a signalling network. The work will be of interest to molecular, developmental and marine biologists and to scientists working on animal metamorphosis.

---

## [Referee Report · Reviewer #1 (Public review)]

Summary:

In this manuscript, the authors use gene functional analysis, pharmacology and live imaging to develop a proposed model of diverse G protein family signalling that takes place in the papillae during the ascidian Ciona larval adhesion to regulate the timing of initiation of the morphological changes of metamorphosis. Their experiments provide solid evidence that antagonistic G protein signalling regulates cAMP levels in the papillae, which provides a threshold for triggering metamorphosis that is reflective of a larva keeping a strong and sustained level of contact with a substrate for a minimum period of approximately half an hour. The authors discuss their reasoning and address different specific aspects of their proposed timing mechanism to provide a logical flow to the manuscript. The results are nicely linked to the ecology of Ciona larval settlement and will be of interest to developmental biologists, neurobiologists, molecular biologists, marine biologists as well as provide information relevant to antifouling and aquaculture sectors.

First, the authors knock down the G proteins Gaq and Gas to show that these genes are important for Ciona larval metamorphosis. They then provide evidence that the Gaq protein acts through a Ca2+ pathway mediated by phospholipase C and inositol triphosphate by showing that inositol phosphate and phospholipase C gene knockdown also inhibits metamorphosis, while overexpression of Gaq or phospholipase C allows larvae to undergo metamorphosis even in the absence of their mechanosensory cue, which is deprived by removing the posterior half of the tail and culturing the larvae on agar-coated dishes. The authors used calcium imaging with a genetically encoded fluorescent calcium sensor to show that Gq knockdown larvae lack a Ca2+ spike in their papillae after mechanostimulation, confirming that Gaq acts through a Ca2+ pathway. Similarly the authors show that overexpression of Gas also enables larvae to metamorphose in the absence of mechanostimulation, suggesting a role for both Gaq and Gas in this process.

To confirm that Gas acts through cAMP signalling, the authors use pharmacological treatment or overexpression of a photoactivating adenylate cyclase to increase cAMP, and show that this also enables larvae to metamorphose in the absence of mechanostimulation, but only when their adhesive papillae are still present. Transcriptome data indicate that both Gs and Gq pathway genes are expressed in the adhesive papillae of the Ciona larva. The authors use a fluorescent cAMP indicator, Pink Flamindo, to show that cAMP increases in the papillae upon adhesion to a substrate, and this increase is lost in Gs and Gq knockdown larvae. Complementary to this, larvae that fail to undergo metamorphosis lack a cAMP increase in papillae.

The authors then provide evidence that GABA signalling within the papillae is acting downstream of the G proteins to induce metamorphosis. Transcriptome data shows that the genes for the GABA-producing enzyme (GAD), and for GABAb receptors, are both expressed in papillae. Pharmacological experiments show that GABA induces metamorphosis in the absence of mechanosensory cues, but only in larvae that retain their papillae. To show that GABA signalling within the papillae, rather than from the brain of the larva is important, the authors also demonstrate that anterior segments of larvae lacking the brain, can also be stimulated to metamorphose by GABA, and show changes in gene expression caused by GABA.

The authors then use a combination of pharmacology and knockdown experiments in the presence or absence of mechanosensory cues to show that Gq/Ca2+ signalling acts upstream of Gs/cAMP signalling. As elevation of cAMP by pharmacology or photoactivating adenylate cyclase rescued GABA pathway mutant larvae, the Gq and Gs pathways were concluded to be downstream of GABA signaling. However, as GABA treatment could still induce Gaq- and Gas-knockdown larvae to metamorphose, suggesting an alternative pathway to metamorphosis, which the authors deduce to be through a third G protein, Gai. They identify an unusual Gai protein that based on transcriptome data is strongly expressed in the papillae. Gai knockdown larvae fail to metamorphose but are rescued by GABA treatment, which can be explained by a potential additional Gai protein being still present (this is confirmed experimentally with MO knockdown experiments). The authors then use overexpression and knockdown experiments to show that the Gai protein acts through Gβγi complex to activate phospholipase C. Their experiments also indicate potential for a complementary or compensatory role for Gai and Gaq signalling through Gβγi. By inhibiting the potassium channel GIRK through knockdown, and the MAPK pathway gene MEK1/2 by pharmacology, the authors also establish a role for these in their proposed model of signalling, allowing GABA and cAMP to compensate or interact with each other.

The strength of this paper is the meticulous and extensive experiments, which are carefully designed to be able to precisely target specific genes in the putative signalling pathway to build step by step a complex model that can demonstrate how metamorphosis of the ascidian larva is timed so as to only occur when strongly attached to a suitable substrate. The unique possibility of inhibiting mechanosensory-induced metamorphosis by removing some of the tail and smoothing the attachment substrate allows the authors to investigate potential effects on both activation and inhibition of metamorphosis, and to confirm that specific signalling pathways are clearly downstream of the initial mechanosensory stimulation. The study is also clear about which aspects of the model still remain unknown, such as which ligands and receptors may be responsible for the binding and activation of Gaq and Gas. Experiments testing metamorphosis of just the anterior region of the larvae nicely demonstrate the need for signalling in the region of the papillae, as do experiments where the papillae are removed, which then block metamorphosis in treatments that would otherwise stimulate it. The final model makes a clear summary of how the extensive experiments all fit together into a cohesive potential signalling network, which can be built upon in the future to potentially integrate the role of sensory cues additional to mechanosensation.

---

## [Referee Report · Reviewer #2 (Public review)]

Summary:

This work aims to characterize the neural signaling cascade underlying the initiation of metamorphosis in Ciona larvae. Combining gene-specific functional analyses, pharmacological experiments, and live imaging approaches, the authors identify the molecular players downstream of GABA to initiate Ciona metamorphosis. The results of this study will serve as a useful framework for future research on animal metamorphosis.

Strengths:

Taking advantage of the Ciona model system, the authors meticulously conducted genetic manipulation and pharmacological experiments to test the epistatic relationships among the signaling players controlling the initiation of Ciona metamorphosis. The experiments were well designed, and the results were convincing. Based on the experimental data, the final working model proposed by the authors will server as an important foundation for further investigation on metamorphosis controls in Ciona and other marine invertebrate larvae.

Weaknesses:

In this revised manuscript, the authors have greatly improved the descriptions of their experimental results, and have clarified my previous concerns. I do not have further comments on "weaknesses".

---

## [Author Response]

The following is the authors’ response to the original reviews

Thank you for your valuable comments, which helped us improve our manuscript. We will make the following modifications in the revised manuscript:

(1) In the first paragraph of the Result section, we will provide a summary of trimeric G proteins in *Ciona* and explain how we focused on Gαs and Gαq in the initial phase of this study.

We added a summary of trimeric G proteins in *Ciona* in the initial part of the Results section (page 6, line 23 to page 8, line 5). In this summary, we added the following sentence explaining the reason we focused on Gas and Gaq in the initial phase of this study: "Among them, we prioritized examining the Gα proteins having an excitatory function (Gαq and Gαs) rather than inhibitory roles since previous studies suggested that excitatory events like Ca^2+^ transient and neuropeptide secretion occur when *Ciona* metamorphose."

(2) As the reviewer 1 suggests, the polymodal roles of papilla neurons are interesting. Although we could not address this through functional analyses in this study, we will add a discussion regarding this aspect. The sentences will be something like the following:

“The recent study (Hoyer et al., 2024) provided several lines of evidence suggesting that PSNs can serve as the sensors of several chemicals in addition to the mechanical stimuli. This finding and our model could be mutually related because these chemicals could modify Ca^2+^ and cAMP production. The use of G protein signaling allows *Ciona* to reflect various environmental stimuli to initiate metamorphosis in the appropriate situation, both mechanically and chemically.”

We added a discussion related to the recent publication by Hoyer and colleagues on page 23, lines 13-18: " A recent study[19] provided several lines of evidence suggesting that PNs can serve as the sensors of several chemicals in addition to mechanical stimuli. This finding and our model could be mutually related because these chemicals could modify Ca^2+^ and cAMP production. G protein signaling allows *Ciona* to reflect various environmental stimuli to initiate metamorphosis either mechanically or chemically according to the situation."

(3) As both reviewers suggested, imaging cAMP on the backgrounds of some G protein knockdowns is essential, and we will conduct the experiments.

We added the data on cAMP imaging in Gas, Gaq, and dvGai_Chr2 knockdown larvae in Supplementary Figure S4C-D and Figure 6E.

(4) We carefully modify the text throughout the manuscript so that the descriptions suitably reflect the results.

We modified the descriptions of experimental results so that the text reflects the results more precisely.

**Reviewer #1:**
Pg1 - need to add an additional '6' to the author list to clarify which two or more authors contributed equally.

We added a 6 as suggested. Thank you for pointing this out.

Pg3 - note that larval adhesive organ applies to not all benthic adults, but to benthic sessile adults this makes it sound like the adhesive organ can trigger metamorphosis but has that been shown? In Ciona or others? Need to specify the role of cells secreting adhesive, vs sensory cells that trigger metamorphosis?

We divided the corresponding sentence into two to clearly state that adhesion and triggering metamorphosis are related but could be different events. Moreover, we modified the sentence to state that physical contact is one example of a cue triggering metamorphosis. We then added another example of a factor triggering metamorphosis—i.e., chemicals from the organisms surrounding the adherence site (page 3, lines 16-20 of the revised version):

"Many marine invertebrates exhibit a benthic lifestyle at the adult stage[4]. Their planktonic larvae have an adhesive organ that secretes adhesives and adheres to a substratum. The cues associated with the adhesion, such as the physical contact with the substratum and a chemical from organisms surrounding the adherence site, can trigger their metamorphosis."

Pg 4 - although mechanosensation is the focus here, could there also be chemoreception/chemoreceptors involved in Ciona metamorphosis? For example, Hoyer et al. 2024 (Current Biology 34(6):1168-1182) concluded that some palp sensory neurons were multimodal and could be both chemo- and mechano-sensory.

We added statements about this recent finding in the Introduction and Discussion sections. In the Introduction (page 4, lines 16-18), however, we also stated that a mechanical stimulus can trigger metamorphosis in the lab without the need to supply these chemicals. This is to emphasize that the mechanical stimulus is the focus of this study. In the Discussion, we added a statement that G-protein signaling could also be used to receive the chemical stimuli (page 23, lines 13-18).

Pg 6 - Before starting functional characterizations, it would be useful to give an overview (table?) of the G proteins found in papillae, and what receptor they are suspected of binding to, or if this is completely unknown, and which downstream pathways they likely activate. That is, to show some results about which G proteins are found in Ciona, and which are found in papillae. In this way, it will make more sense for readers when the Gai is suddenly introduced later, following the sections of Gaq and Gas.

Thank you for your idea to improve the readability of this manuscript. In the initial part of the Results section (page 6, line 22 to page 8, line 5), we added descriptions of the repertoire of trimeric G-proteins in *Ciona*, including phylogenetic analyses, and expression in the papillae based on RNA-seq data, followed by the reason why we initially focused on Gaq and Gas. The data are displayed in Supplementary Figure S1. The phylogenetic analyses were modified from those shown in Supplementary Figure S5 of the previous version. We also added the general downstream activities of Gas, Gai and Gaq in the Introduction section (page 6, lines 10-12). Considering the contents, the general function of Ga12/13 was stated in the Results section (page 8, lines 2-3).

We did not add the information about their partner receptors in this early section. This is because there are many candidates, and we could not pick some of them. Instead, we described our current suppositions about their possible partners in the Discussion (page 23, line 22 to page 24, line 19). However, we suspect that there are more candidates, and we wish to promote unbiased research in the future.

Pg 9 - would be good to know the timing of this PF fluorescence increase and the timing of stimulation in the text here, relevant to the 30-min gap before metamorphosis initiation

We added the start times for the cAMP reduction and re-upregulation in the following sentence (page 11, lines 17-18): "The cAMP reduction and increase respectively started at 35 seconds and 4 min 40 seconds after stimulation on average."

Pg 28 - Phylogenetic analysis: Given that the results may be of interest to metamorphosis in other marine invertebrates as discussed in the last paragraph of the paper, it would be useful to include G proteins from these other animal phyla where available in the phylogenetic tree. Similarly, in Figure S5A it would be useful to highlight further all the different Ciona G proteins, and the different protein families, through the use of additional colour/labelling (regardless of whether this remains Fig S5A, or becomes part of the main figures)

We drew a phylogenetic tree of G-proteins including those in some sessile and benthic animals (barnacle, sea anemone, hydra, sponge, sea urchin and shell). However, we decided not to add the tree in the revised version because, unfortunately, the bootstrap values of many branches were not high enough to have confidence in the results. We hope you understand our decision. *Ciona* divergent G-proteins are likely to be specific to *Ciona*.

According to your comment, we highlighted all *Ciona* G alpha proteins in red in Figure S5A, which is now Figure S1A in the revised version.

Figure 3E and Figure S3 - is the data shown as an average of all larvae measured (n=5 and n=4) or is it data from one representative larva out of the 4-5 measured? This needs clarification.

The original graphs in Figure 3E and Figure S3 are typical examples. We added the graphs summarizing data of all larvae in each experimental condition in Supplementary Figure S4 (corresponding to Supplementary Figure S3 of the original version). Figure 3E remains as a typical example of the result of a single larva to explain our data analysis in detail.

Experimental suggestion - As mentioned above, one missing detail seems to be the need for evidence that cAMP is elevated in the papillae directly as a result of Gs activation- this could be shown with measurement of cAMP via PF in Gs knockdown larvae that are mechanically stimulated compared to wildtype stimulated and non-stimulated?

Thank you for your suggestion. The experiments are indeed important. We added the data of Pink Flamindo imaging in the *Gas*, *Gaq* and *dvGai_Chr2* knockdown conditions. The results of *Gas* and *Gaq* knockdowns are described in page 11, line 24 to page 12, line 5, and are displayed in Supplementary Figure S4C-D. The result of *dvGai_Chr2* knockdown is given on page 16, lines 20-22 and shown in Figure 6E.

In order to insert the data of cAMP imaging of *dvGai_Chr2* knockdown larvae, we transferred some panels of Figure 6 to Supplementary Figure S6. In addition, the knockdown data of *dvGαi_Chr4* and double knockdowns of *Gai* genes are also included in Supplementary Figure S6.

**Reviewer #2:**
Page 6, line 3-4 in the first paragraph of the "Results"; the authors state "Neither morphant showed any signature of metamorphosis even though both were allowed to adhere to the base of culture dishes...". However, judging from Fig. 1E, "the percentage of metamorphosis initiation" (indicated by the initiation of tail regression) in Gαq morphans is not close to 0 (average about 40%), thus I am not convinced this observation can be described as "Neither morphant showed any signature of metamorphosis..." in this sentence.

Thank you for your suggestion. In writing the original text, we oversimplified some of the descriptions when trying to improve the readability. We agree this resulted in imprecision in places. We have revised all these passages in our revision. In this particular case, we softened the overly emphatic statement to better reflect the results, changing “... any signature of metamorphosis...” to “... reduced rate of metamorphosis initiation...” In addition, we stated that the effect of *G*α*q* MO was weaker than that of *G*α*s* MO on page 8, lines 10-12. The weaker effect of *Gaq* MO was due to the redundant role of the Gi pathway, which is shown on page 17, lines 10-17, and in Figure 6G-H.

Similarly, in the next paragraph describing the knockdown of PLCβ1/2/3, PLCβ4, and IP3R genes, the authors appear to neglect there is a weaker effect of the PLCβ4 MO, and simply described the results as "The knockdown larvae of these three genes failed to start metamorphosis". Based on Fig. 1H, about 30% of the PLCβ4 MO-injected animals still initiated tail regeneration. This difference may have some biological meanings and thus should be described more precisely.

We added the following sentence on page 8, lines 18-19 of the revised version: “The effect of *PLCβ4* MO was weaker than those of the other MOs, suggesting that this PLC plays an auxiliary role.”

Page 7, second paragraph, on the description of GCaMP8 fluorescence and also at the end of Fig. 1O legend, the citation to "Figure S1" is confusing; Fig. S1 is the phylogenetic tree of PLCβ proteins. Is there additional data regarding this Gαq MO plus GCaMP8 mRNA injection experiment?

Figure S1 of the original version corresponds to Figure S2 of the revised version. To avoid confusion, we deleted this citation from the legend of Figure 1O. By this modification, the sentence stating the repertoire of PLCb and IP3R in Ciona (page 8, lines 15-16) is the only sentence citing Figure S2 in the revised version.

Page 8, first sentence; The purpose of theophylline treatment is not to prevent larvae from adhesion, thus I would suggest modifying this sentence to: "We treated wild-type larvae with theophylline after tail amputation, and we observed that most theophylline-treated larvae completed tail regression without adhesion (Figure 2D-F)".

We modified the sentence according to your comment. Thank you for your suggestion.

Page 9, second paragraph; judging from the data presented in Fig. 3C, I think this description: "when papillae were removed from larvae, theophylline failed to induce metamorphosis" is not accurate, because about ~30% of the Papilla cut +Theophylline-treated larvae still initiated their tail regression. This needs to be explained clearly.

We modified the sentence (page 11, lines 2-3) as follows: “...the average rate of metamorphosis induction by theophylline was reduced from 100% to 30%...”

Similarly in the next few sentences regarding the results presented in Fig, 3D, the effects of overexpressing those genes are not uniform. While amputation of papillae in larvae overexpressing caPLCβ1/2/3 could inhibit metamorphosis almost completely, papilla cut seems to have a weaker effect on caGαq, caGαs, and bPAC-overexpressing larvae.

We added a description explaining that ca*PLCβ1/2/3* was the most sensitive to papilla amputation, and the possibility that PLCβ1/2/3 works specifically in the papillae (page 11, lines 9-11): “Among these experiments, ca*PLCβ1/2/3* overexpression was the most sensitive to papilla amputation, suggesting that PLCβ1/2/3 acts specifically in the papillae during metamorphosis.”

Page 9, the paragraph on using the fluorescent cAMP indicator; there is a discrepancy between the described developmental time when the authors conducted this experiment and the metamorphosis competent timing (after 24hpf) described on page 7. On page 26, the authors describe "The Pink Flamindo mRNA-injected larvae were immobilized on Poly L lysine-coated glass bottom dishes at 20-21 hpf...". Did the authors start stimulating the larvae to observe the fluorescent signal soon after immobilization, or wait several hours until the larvae passed 24hpf and then conduct the experiment?

The latter is the case. The immobilized larvae were kept until they acquired the competence for metamorphosis and then stimulation/recording was carried out. This point is described in the Materials and Methods section of the revised version (page 29, lines 16-18):

"The Pink Flamindo mRNA-injected larvae were immobilized on Poly L lysine-coated glass-bottom dishes at 20-21 hpf, and stimulated their adhesive papillae around 25 hpf."

Page 10, the description "...Gαq morphants initiated metamorphosis when caGαs was overexpressed in the nervous system (Figure 4F)". It should be noted that the result is only a partial rescue. To be precise, this description needs to be modified.

We changed the sentence to reflect the results more precisely (page 14, lines 2-3): “Moreover, ca*Gαs* overexpression in the nervous system significantly, although not perfectly, ameliorated the effect of *Gαq* MO (Figure 4F).”

Page 12-13, This description and the figure 5E presented is a bit confusing to me. The figure legend for 5E: "GABA is necessary for Ca2+ transient in the adhesive papillae (arrow)" But the arrow in this image points to a place with no fluorescent signal, and on the upper corner it labeled as "29% (n=17)". Does that mean the proportion of "no Ca2+ increase after stimulation" was 29% among the 17 samples examined? Or actually, is the other way around that 81% of the examined larvae did not show Ca2+ signal increase after stimulation?

The latter is the case. We added a caption explaining this clearly in the Figure legend: “The percentage and number exhibit the rate of animals showing Ca^2+^ transient in the papillae.”

Page 13, second paragraph; I do not agree with the overly simplified description that "GABA significantly ameliorated the metamorphosis-failed phenocopies of Gαq, PLCβ, and Gαs morphants". As shown in Fig. 5F-H, adding GABA exerts different levels of partial rescue effect on each morphant, and thus should be described clearly.

When the outliers are neglected, the effect of GABA is most evident in *Gαs* knockdowns. This suggests that the target(s) of GABA signaling is more likely to be Gq pathway components. We added the following sentence to the revised version (page 15, lines 14-16):

“Among the three morphants, GABA exhibited the most effective rescues in *Gαs* knockdowns than *Gαq* and *PLCβ*.”

In addition, we think this sentence establishes a more logical connection with the sentence that follows it: “These results could be explained by assuming enhancement of the Gq pathway by GABA through PLCβ and another GABA-mediated metamorphic pathway bypassing Gq components.” Thank you for your suggestion.

The section "Contribution of Gi to metamorphosis" confirmed the possibility that GABA signaling targets Gq pathway components.

Page 13, the first paragraph on "Contribution of Gi to metamorphosis"; the description that "The knockdown of this gene (Gαi) exhibited a significantly reduced rate of metamorphosis;..." is misleading. I would suggest modifying the entire sentence as "The knockdown of this gene (Gαi) exhibited a moderate (although statistically significant) reduction of metamorphosis rate, suggesting the presence of another Gαi regulating metamorphosis".

Thank you for your suggestion. We modified the sentence (page 16, lines 2-4 in the revised version) as recommended. We believe the description is much improved.

Page 20, the last sentence about Ciona papilla neurons expressing transcription factor Islet; the authors seem to attempt to make some comparison with the vertebrate pancreatic beta cells in this paragraph, but the comparison and the argument are not fully developed in this current format.

To deepen this discussion, we added the following sentence (page 23, lines 10-12): “The atypical secretion of GABA might depend on the transcription factor like Islet shared between *Ciona* papilla neurons and vertebrate beta cells.”

However, we would like to limit the depth of our discussion on this point, as we hope to expand on it further in future studies.

Other suggestions:Page 3, second paragraph: as they become unable to "move" after metamorphosis -> "relocate"

We corrected the word as suggested.

Page 4, second paragraph: In the first sentence, the author states the current understanding of chordate phylogeny and cites Delsuc et al. 2006 Nature paper at the end of this sentence. However, in this paper cephalochordates were erroneously grouped with echinoderms, and thus chordates did not form a monophyletic clade. A later paper by Bourlat et al, (Nature 444:85-88, 2006) corrected this problem, and subsequently Dulsuc et al. also published another paper (genesis, 46:592-604, 2008) with broader sampling to overcome this problem. These later publications need to be included for the sake of correctness.

We added this reference.

Page 14, regarding the redundant function of the typical Gαi protein in the papillae; the authors may try double KD of Gαi and dvGαi_Chr2 in their experimental system to test this idea.

We carried out double knockdown of typical Gai and dvGαi_Chr2. However, we could not address their redundant role sufficiently because most of the double knockdown larvae exhibited severe shape malformation.

dvGαi_Chr4 is also expressed in the papillae. We carried out knockdown of this gene, to find that the knockdown resulted in very minor but statistically significant reduction of the metamorphosis rate, suggesting that this Gai also plays a supportive role in metamorphosis. We also carried out double knockdown of dvGαi_Chr2 and dvGαi_Chr4. The double KD larvae exhibited responsiveness to GABA, probably because of the presence of typical Gai.

These results are described on page 16, lines 2-18, and the data are shown in Supplementary Figure S6A-D of the revised version.

Responses to the Reviewing editor's comments:

"Larvae of the ascidian Ciona initiate metamorphosis tens of minutes after adhesion to a substratum via its adhesive organ." - Larvae is plural so change to 'via their adhesive organ'

The sentence was corrected as suggested.

"Metamorphosis is a widespread feature of animal development that allows them" - revise the sentence, e.g. "Metamorphosis is a widespread feature of development that allows animals"

The sentence was corrected as suggested.

"GABA synthase (GAD)" GAD is not called GABA synthase but glutamate decarboxylase - clarify, e.g. encoding the enzyme synthesizing GABA called glutamate decarboxylase (GAD)

This part was corrected exactly as suggested. Thank you.

"IP3 is received by its receptor on the endoplasmic reticulum (ER) and releases calcium ion (Ca2+)" revise to "IP3 is received by its receptor on the endoplasmic reticulum (ER) that releases calcium ion (Ca2+)"

The sentence was corrected as suggested.

"Moreover, GPCR is implicated as the mediator of settlement" - GPCRs are implicated

This sentence was modified as suggested.